# Mechanically activated piezo channels modulate outflow tract valve development through the Yap1 and Klf2-Notch signaling axis

Anne-Laure Duchemin[1,2,3,4], Hélène Vignes[1,2,3,4], Julien Vermot[1,2,3,4]*

[1]Institut de Génétique et de Biologie Moléculaire et Cellulaire, Illkirch, France; [2]Centre National de la Recherche Scientifique, Illkirch, France; [3]Institut National de la Santé et de la Recherche Médicale, Illkirch, France; [4]Université de Strasbourg, Illkirch, France

**Abstract** Mechanical forces are well known for modulating heart valve developmental programs. Yet, it is still unclear how genetic programs and mechanosensation interact during heart valve development. Here, we assessed the mechanosensitive pathways involved during zebrafish outflow tract (OFT) valve development in vivo. Our results show that the hippo effector Yap1, Klf2, and the Notch signaling pathway are all essential for OFT valve morphogenesis in response to mechanical forces, albeit active in different cell layers. Furthermore, we show that Piezo and TRP mechanosensitive channels are important factors modulating these pathways. In addition, live reporters reveal that Piezo controls Klf2 and Notch activity in the endothelium and Yap1 localization in the smooth muscle progenitors to coordinate OFT valve morphogenesis. Together, this work identifies a unique morphogenetic program during OFT valve formation and places Piezo as a central modulator of the cell response to forces in this process.
DOI: https://doi.org/10.7554/eLife.44706.001

*For correspondence:
julien@igbmc.fr

**Competing interests:** The authors declare that no competing interests exist.

## Introduction

Heart pumping and shaping take place concomitantly during embryonic development. These two processes require a tight and dynamic coordination between mechanical forces and tissue morphogenesis. Heart valve development is a great model for studying these interactions when considering that heart valve defects are common congenital cardiac malformation in human (*Hoffman and Kaplan, 2002*; *Øyen et al., 2009*). The four-chambered heart contains two different sets of valves (*Lin et al., 2012*):

   - Tricuspide valves including the two semilunar (SL) valves, the aortic valve and the pulmonary valve, as well as the tricuspide valve located between the right atrium and the right ventricle.

   - A bicuspide valve called the mitral valve separating the left atrium from the left ventricle. Abnormalities of the arterial valve leaflets are the most-common congenital malformations, in particular bicuspid aortic valve (*Hoffman and Kaplan, 2002*). Aortic valves mainly derive from endocardial cushion progenitors with a potential contribution from other cellular sources (epicardial cells and neural crest) (*Wu et al., 2017*). Generally, valve formation depends on two main events: cell proliferation, which is mainly mediated by the *vegf-nfat1* pathway, matrix deposition and an endothelial-to-mesenchymal transformation (endMT) under the control of Gata transcription factors (*Laforest et al., 2011*; *Stefanovic et al., 2014*), Notch signaling (*Luxán et al., 2016*), *Smad/tgf-beta/Bmp*, and *Wnt-beta catenin* signals (*Combs and Yutzey, 2009*). Nevertheless, recent evidence

suggests that arterial valves develop differently from atrioventricular valves by differentiating directly from progenitors in the outflow wall independently from endMT in mouse (*Eley et al., 2018*).

Congenital valve defects may originate from developmental origins and/or abnormal haemodynamic forces between these two sets of valves, and it remains unclear how general these developmental programs are. Aortic valves are located in areas of high flow velocity and mechanical forces have a great impact on valve morphogenesis (*Butcher et al., 2008*). Abnormal blood circulation is widely recognized as a cardiovascular risk factor and abnormal mechanotransduction has been associated with valvulopathies (*Bäck et al., 2013*). Congenital heart valve malformations are usually associated with genetic mutations in genes essential for heart valve development, such as signaling factors (Notch1, TGFβ) for the aortic valves (*Bäck et al., 2013*), and actin-binding proteins (FilaminA) for the mitral valves (*Sauls et al., 2012*). The reoccurring discovery of genetic mutations linking valve defects with genes involved in controlling developmental programs (e.g., in NOTCH1, TBX5, GATA4, TBX20, LMCD1, TNS1, and DCHS1) (*PROMESA investigators et al., 2015*; *Durst et al., 2015*; *Garg et al., 2005*; *Richards and Garg, 2010*), has spurred interest in valve morphogenesis. A key issue is to further define the genetic or environmental causes of valve malformation.

The zebrafish constitutes a powerful model to study cardiac valve development and the role of mechanical forces at the cellular scale. Zebrafish heart is two chambered and contains three sets of valves (the outflow tract (OFT), atrioventricular (AVC) and the inflow tract (IFT) valve [*Figure 1A*]) that are all bicuspid (*Beis et al., 2005*; *Hsu et al., 2019*; *Tessadori et al., 2012*). While the developmental programs driving mitral valve development in response to mechanical forces start to be well established in zebrafish, less is known about OFT and IFT valves (*Paolini and Abdelilah-Seyfried, 2018*; *Steed et al., 2016a*). The cellular processes leading to valve formation are dynamic and are particularly challenging to address in vivo. Zebrafish heart valves originate from progenitors located in the ventricle and atrium that generate the valve leaflets through a coordinated set of endocardial tissue movements (*Boselli et al., 2017*; *Pestel et al., 2016*; *Steed et al., 2016a*; *Steed et al., 2016b*; *Vermot et al., 2009*). The sequence of cellular events leading to AVC valve formation in zebrafish embryonic hearts is initiated through cell shape changes that lead to EC convergence towards the AVC (*Boselli et al., 2017*) and cellular rearrangements that will form a multilayered tissue (*Beis et al., 2005*; *Pestel et al., 2016*; *Scherz et al., 2008*; *Steed et al., 2016b*). In the zebrafish AVC, blood flow and Klf2a control *notch1b* and *bmp4* expression, both of which are necessary for valve formation (*Vermot et al., 2009*). Klf2a regulates the deposition of matrix protein (in particular Fibronectin1) in the valve forming area (*Steed et al., 2016b*), as well as Wnt signaling by controlling *wnt9b* expression (*Goddard et al., 2017*). The latter is consistent with the fact that canonical Wnt signals arise specifically in sub-endocardial, abluminal cells and that these Wnt signals are dependent upon hemodynamic forces in zebrafish (*Pestel et al., 2016*). In addition, Notch signaling is essential for aortic valve formation (*Garg, 2016*) and OFT development (*MacGrogan et al., 2016*; *Wang et al., 2017*). The role of mechanical forces during OFT valve development at the cellular and molecular scale, however, is largely unknown.

Fluid shear stress is an important environmental cue that governs vascular physiology and pathology (*Baeyens et al., 2016*), but the molecular mechanisms that mediate endocardial responses to flow are only partially understood. In zebrafish, the mechanosensitive channels Trpv4 and Trpp2 modulate endocardial calcium signaling and *klf2a* expression is necessary for valve morphogenesis and downstream pathway activation (*Heckel et al., 2015*; *Steed et al., 2016b*). Notch signaling is tightly involved in cellular mechanosensitive responses in human aortic valves (*Godby et al., 2014*) and Notch1 is a potent mechanosensor in adult arteries (*Mack et al., 2017*). More recently, it has been shown that stretch-sensitive channels from the Piezo family (*Murthy et al., 2017*) are important for vascular development (*Li et al., 2014*; *Ranade et al., 2014*) and lymphatic valve formation (*Nonomura et al., 2018*). In the embryo, Piezo channels exert essential roles during cell differentiation (*He et al., 2018*) and can affect lineage choice by modulating the nuclear localization of the mechanoreactive transcription coactivator Yap (*Pathak et al., 2014*). Nevertheless, the role of Piezo-mediated mechanotransduction during cardiac development and its potential targets remain unclear.

In the developing cardiovascular system, biomechanics is key for modulating flow propagation (*Anton et al., 2013*). In the teleost heart, the OFT constitutes a specialized organ comprising the conus arteriosus (CA) and the bulbus arteriosus (BA) (*Figure 1A*). The BA dampens the pressure wave down the arterial tree (*Braun et al., 2003b*). To perform its function, the BA expresses elastic

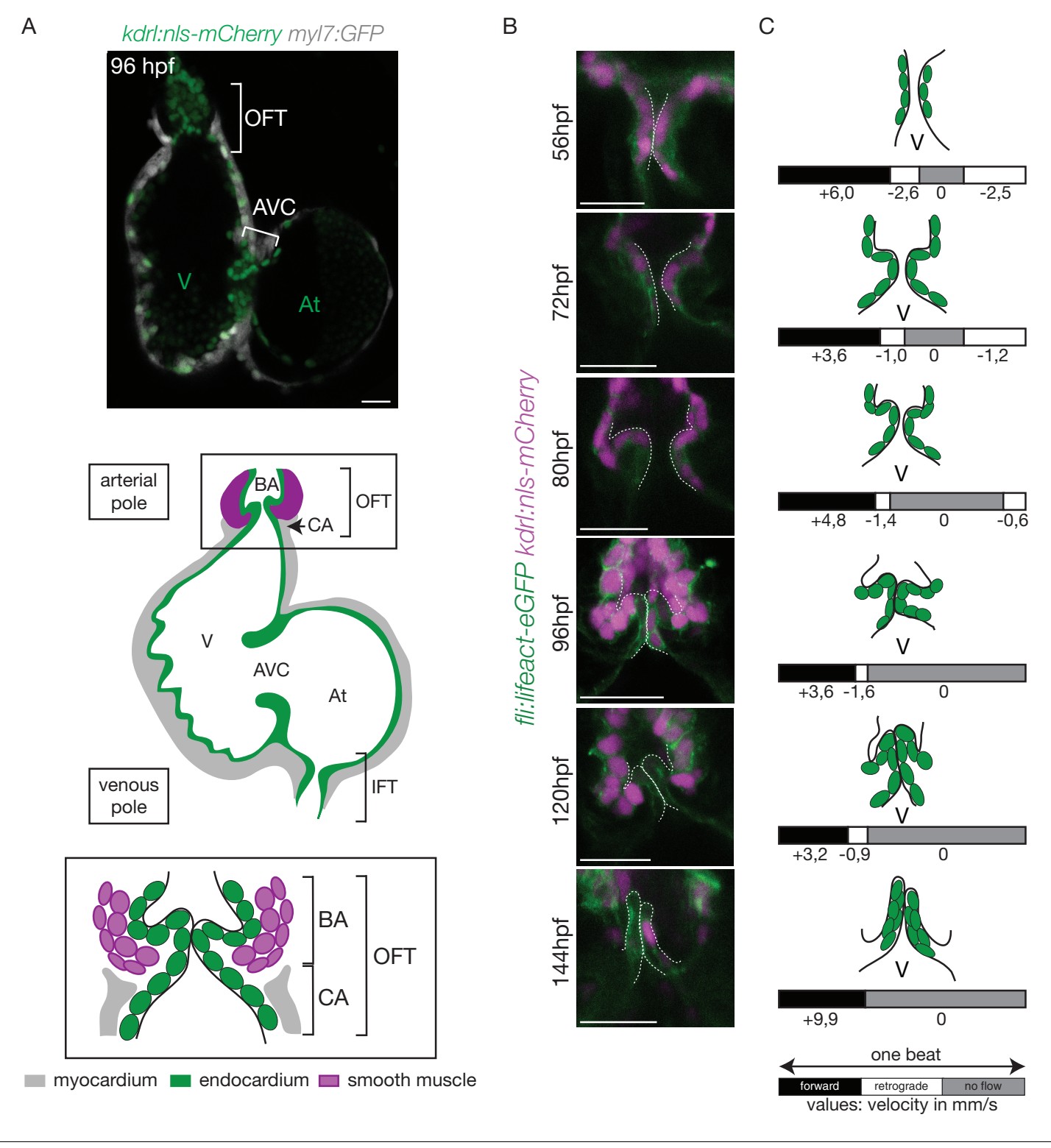

**Figure 1.** The OFT develops from 56 hpf to form functional valves at 144 hpf. (**A**) Top: Z-section of the double transgenic line *Tg(kdrl:nls-mCherry; myl7:GFP)* showing the overall structure of the heart. Bottom: Scheme of the zebrafish heart with the endocardium, myocardium and smooth muscles and zoom on the OFT structure. The OFT includes the CA and the BA. The CA is the zone of the myocardial connection of the ventricle to the BA. The BA is after the ventricle and is surrounded by smooth muscles. Scheme adapted from *Felker et al. (2018)*. OFT: outflow tract, IFT: inflow tract, AVC: atrioventricular canal, At: atrium, V: ventricle, BA: bulbus arteriosus, CA: conus arteriosus. Scale bar: 20 μm. (**B**) Z-sections of the double transgenic line
*Figure 1 continued on next page*

*Figure 1 continued*

*Tg(fli:lifeact-EGFP; kdrl:nls-mCherry)* at different time-points showing the endocardial OFT structure. Scale bar: 25 μm. (C) Schematic representation summarizing the formation of the valve leaflets over time and flow profile in the OFT during development (from 56 hpf to 144 hpf) showing the forward flow (black), retrograde flow (white) and no flow (grey) fractions with the velocity of the red blood cells (in mm/s) using the double transgenic line *Tg (gata1:ds-red; kdrl:EGFP)*. V: ventricle.

DOI: https://doi.org/10.7554/eLife.44706.002

fiber genes that are thought to provide the mechanical properties necessary for its physiological function (*Braun et al., 2003a*; *Braun et al., 2003b*; *Keith et al., 1977*). The BA is separated from the ventricle by the OFT valve and is composed of smooth muscle. The extracellular matrix (ECM) gene, *elastin b*, contributes to the development of the BA by regulating cell fate determination of cardiac precursor cells into smooth muscle via a process that involves the mechanotransducer Yap1 (*Moriyama et al., 2016*). How these factors contribute to OFT valve development and interact with other mechanosensitive pathways remains unclear.

In this study, we investigated the signaling events taking place during OFT valve formation and addressed their regulation by the mechanosensitive channels Piezo and transient potential channels (Trp) as well as the flow-responsive transcription factor Klf2a. We show that OFT valve formation proceeds via an initial stage of endothelial cell folding, which is associated with the generation of a cluster of smooth muscle cell progenitors surrounding the endothelial layer. Subsequent global tissue remodeling events result in the appearance of functional leaflets, which defines a unique process of valvulogenesis. Using live reporters to highlight the signaling changes accompanying these temporally coordinated cell-movement events and genetics, we identified Notch and Klf2 as key flow-dependent factors as well as Yap1 as necessary factors for the correct coordination of OFT valvulogenesis. We show that Piezo and Trp channels are key regulators of *klf2* activity in the endothelium and that piezo modulates Yap1 localization in the smooth muscle cells, providing a molecular link between mechanosensitivity and cell signaling in the multilayered valve structure. These data describe the cell responses that are coordinated by the mechanical environment and mechanotransduction via mechanosensitive channels in the endothelium.

## Results

### Outflow tract valve morphogenesis is unique

In order to better understand the roles played by blood flow during outflow tract (OFT) valve development, we have developed imaging techniques to capture cardiac motion and analyze blood flow in the OFT. Live imaging of the double transgenic line *Tg(gata1:ds-red; kdrl:EGFP)* to follow red blood cells and endothelial cell wall movements reveal dramatic changes in intracardiac blood flow patterns during OFT valve development: as the heart matures, blood flow in the OFT is bidirectional until functional valve leaflets emerge in the OFT at 144 hpf (*Figure 1A–C*). Throughout development, the periods within the cardiac cycle in which reversing flow can be observed decrease in length until 144 hpf, the stage at which we could not observe reversing flow anymore (*Figure 1C*). Using the *Tg(fli:lifeact-EGFP; kdrl:nls-mCherry)* line, which labels endothelial cells, we found that these flow profile modifications are linked to changes in OFT tissue geometry and the state of OFT maturation (*Figure 1B*). At 56 hpf, the endothelium resembles a tube (*Figure 1B,C*), maturing into cushions by 72 hpf, into premature leaflets by 96 hpf and finally into elongated, thin leaflets by 144 hpf (*Figure 1B,C*).

To better characterize how the endothelium changes shape over time and how cells reorganize to form OFT valves, we performed photoconversion experiments using the *Tg (fli:gal4FF;UAS:kaede)* (*Figure 2A*). We photoconverted Kaede from green to red in the cells located in the anterior, middle or posterior part of the valve at 72 hpf and assessed their position at 96 hpf and 120 hpf (*Figure 2A, B*). The results suggest that the endothelium folds to form the valve without multilayering (*Figure 2B,C* and *Figure 2—figure supplement 1A*). Indeed, we could observe that the photoconverted cells remain attached to each other and do not show signs of delamination as observed in the AVC (*Figure 2B,C* and *Figure 2—figure supplement 1B*).

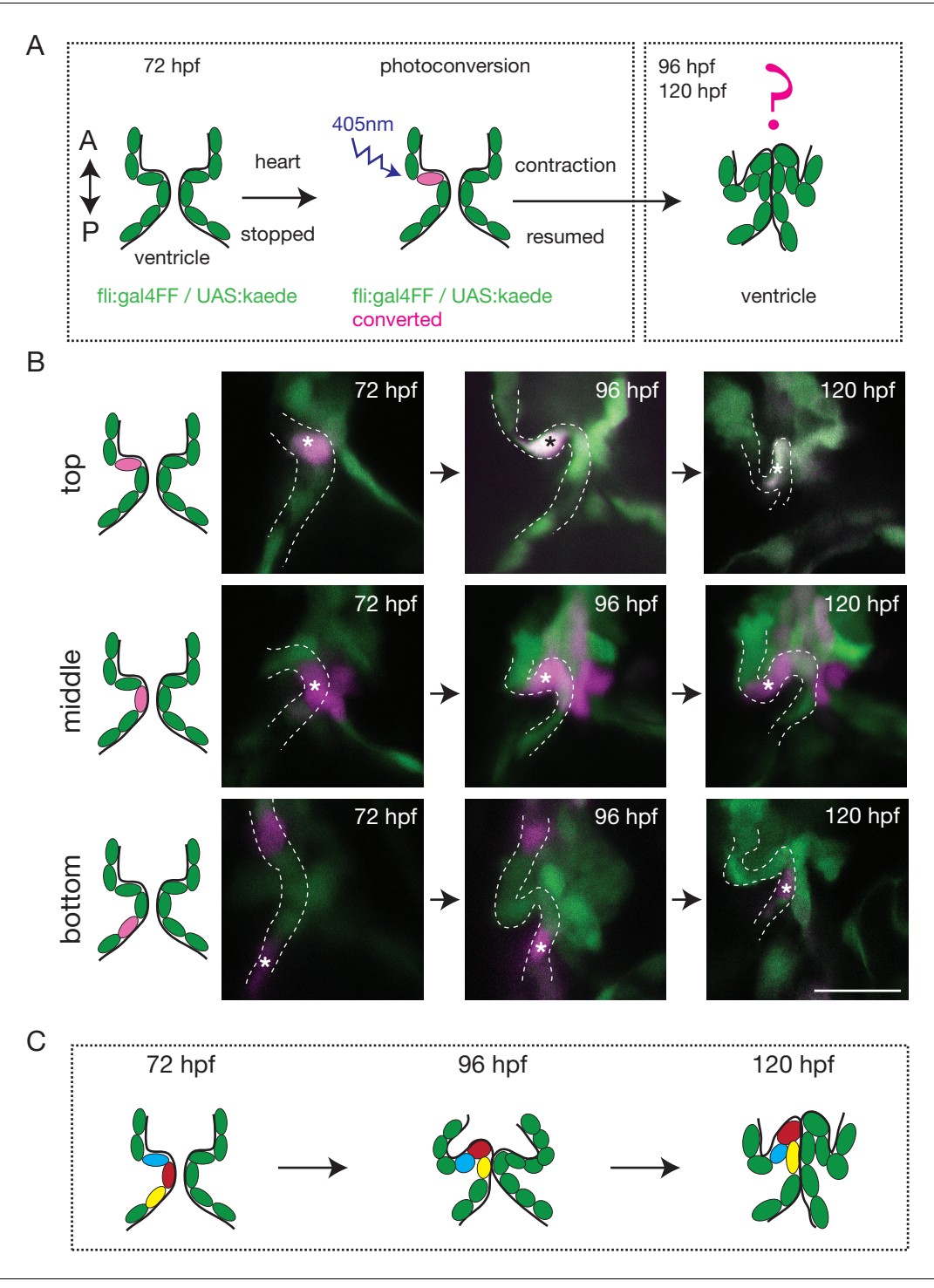

**Figure 2.** The endothelium contribution to emerging OFT valve leaflets. (A) Experimental set-up for the photoconversion studies. Heart was stopped at 72 hpf, the region of interest exposed to 405 nm light to convert kaede from green to red (shown in magenta) fluorescent form and heart contraction was resumed until 120 hpf. Beating hearts were imaged at 96 hpf and 120 hpf by spinning disk microscopy. A. Anterior, P. Posterior. (B) Z-sections of the *Tg(fli:gal4FF; UAS:Kaede)* line just after photoconversion (72 hpf), and at 96 hpf and 120 hpf. The star highlights the photoconverted cell in the top (n = 6), middle (n = 5) and bottom (n = 4) part of the OFT valve. The other photoconverted cell anteriorly goes out of the frame. Scale bar: 20 μm. Results obtained from three independent experiments. (C) Schematic representation of the results of the photoconversion studies showing the folding of the endothelium in the OFT.

*Figure 2 continued on next page*

*Figure 2 continued*

DOI: https://doi.org/10.7554/eLife.44706.003

The following figure supplement is available for figure 2:

**Figure supplement 1.** The endothelium contribution to emerging OFT versus AVC valves.

DOI: https://doi.org/10.7554/eLife.44706.004

Together, these results suggest that OFT valves form by a folding process that might involve the adjacent tissue.

The cellular contribution of the OFT valve led us to the hypothesis that the surrounding tissue could contribute to valve morphogenesis. We analyzed Fibronectin1 (Fn1) expression in the OFT by counterstaining the *Tg(kdrl:EGFP)* with the Fn1 antibody (*Figure 3A–D*). We found that Fn1 deposition in the OFT is different from that in the AVC (*Steed et al., 2016a*). At 72 hpf, cushions appear and Fn1 is observed in the cells around the endothelial cells, that are themselves surrounded at their base by myocardium in the posterior part (*Figure 3A,B,C,D,E*). Fn1 expression level increases at 96 hpf and delineates a group of cells surrounding the OFT as well as in the basal side of a few endothelial cells that form the cushions (*Figure 3C*, arrows, E). In the developed leaflets at 120 hpf, Fn1 expression is maintained within the leaflets (*Figure 3D,E*). We found that most of the cells surrounding the endothelium expressing Fn1 also express Elastin b (Elnb, Eln2 or Tropoelastin), a marker of smooth muscle cells (*Grimes et al., 2006*; *Miao et al., 2007*; *Paffett-Lugassy et al., 2017*) (*Figure 3B', C', D'*). We could confirm that the cells surrounding the Fn1 cells are not myocardial cells as the myocardium stops just before the BA region (*Figure 3A,E*). To better characterize the smooth muscle identity and their activity, we performed a DAF-FMDA assay and counterstained with Fn1 at 72 hpf, 96 hpf and 120 hpf (*Figure 3—figure supplement 1A*). These results suggest that all cells expressing Fn1 are also DAF-FMDA-positive (*Figure 3—figure supplement 1A*). Some of these cells also express *Tg(wt1a:GFP)* (asterisks in *Figure 3—figure supplement 1B*) a marker of epicardial cells (*Peralta et al., 2013*). One hypothesis could therefore be that they might have originated from epicardial precursors. Together, these results suggest a developmental sequence of OFT morphogenesis in vivo where endothelial cell reorganization is associated with changes in gene expression in the surrounding smooth muscle cell progenitors. This indicates that the OFT morphogenesis involves remodeling of not just the endothelium, but also of a group of smooth muscle cells that express Fn1 and Elnb, and are functionally active.

We conclude that the tissue remodeling occurring during OFT valve development is significantly different from AVC valve development where the endocardium is the main remodeling tissue (*Beis et al., 2005*; *Pestel et al., 2016*; *Steed et al., 2016b*).

## Klf2, Notch signaling, and Hippo pathways are active in the OFT in different cell layers and are all necessary for proper OFT valve development

To elucidate how these early events are regulated, we sought to determine the mechanosensitive signaling pathways activated at these early stages of OFT valve formation. We first assessed the activity of a Klf2a reporter line (*Tg(klf2a:H2B-GFP)*) (*Figure 4A*), which is a well described flow responsive reporter (*Heckel et al., 2015*; *Steed et al., 2016b*) and the Notch reporter *Tg(tp1:dGFP)* (*Figure 4B*) which is well active in the progenitors of the AVC cardiac valves (*Pestel et al., 2016*). We could observe a specific activation of the Klf2a reporter in the OFT endothelium (*Figure 4A,C* and *Figure 4—figure supplement 1*), in particular in the ventricular part of the valve (posterior) from 72 hpf to 120 hpf (*Figure 4C,D* and *Figure 4—figure supplement 1*). Similarly, the Notch reporter *Tg(tp1:dGFP)* is specifically expressed in the OFT endothelium (*Figure 4B,C*), in the ventricular part of the valve from 72 hpf to 120 hpf (*Figure 4C,D*). Interestingly, the spatial activation of the reporter varies within the valve forming area - the transgene activation is stronger in the posterior part of the valve than the anterior part of the valve throughout the process of valve maturation (*Figure 4C,D*). We made similar observations for the Notch reporter (*Figure 4C,D*). These results suggest that Klf2a and Notch signaling are activated specifically in the part of the valve corresponding to where the OFT has the smallest diameter and where shear stress is expected to be the highest, consistent with the hypothesis that Klf2a and Notch signaling are flow-responsive.

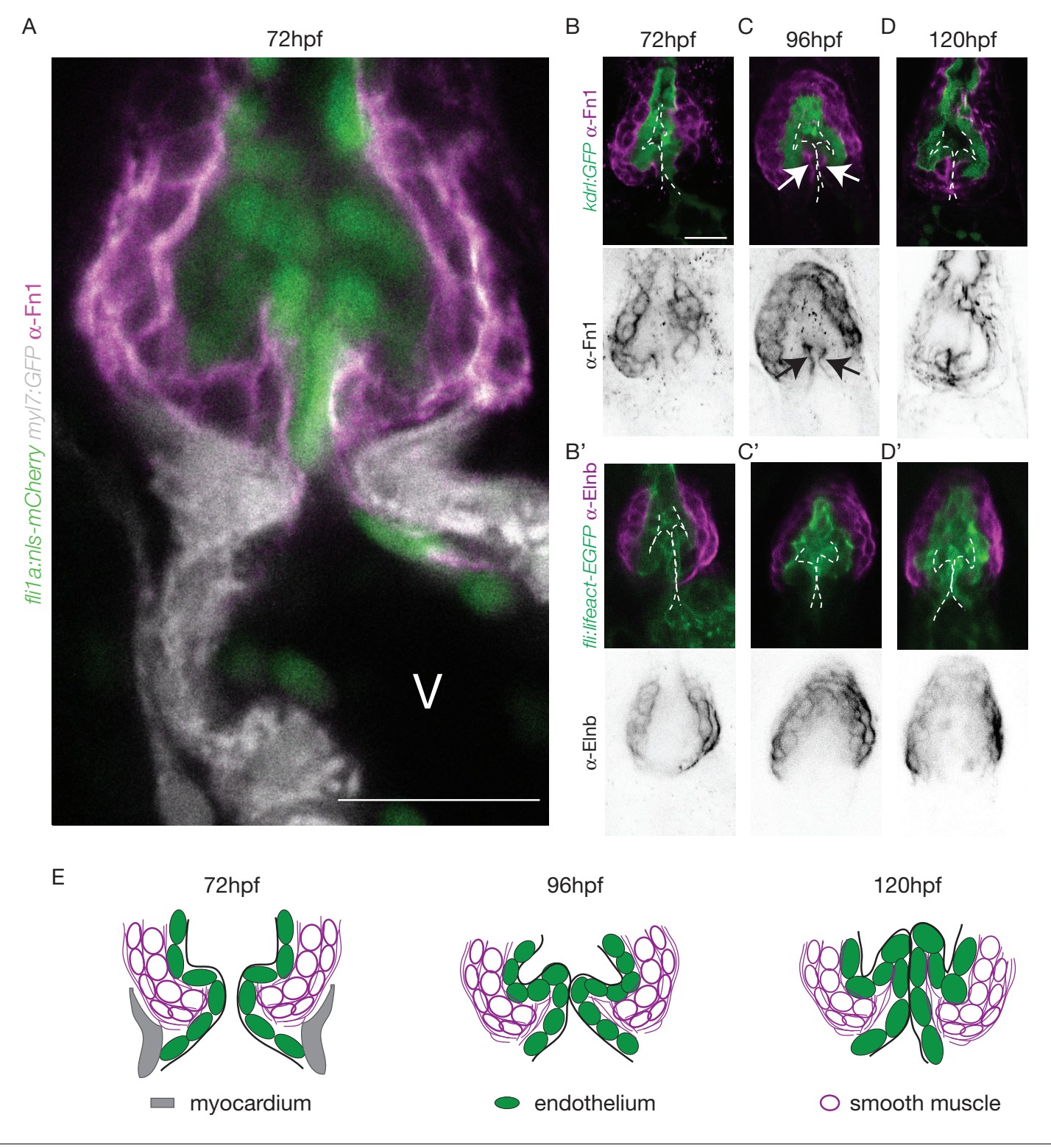

**Figure 3.** The OFT endothelium is surrounded by smooth muscle cell progenitors expressing fibronectin and elastin. (**A**) Staining of Fibronectin1 (magenta) on *Tg(myl7:GFP; fli1a:nls-mCherry)*, highlighting the myocardium (white) and the endothelium (green) at 72 hpf. Scale bar: 20 µm. V: ventricle. Fibronectin1 (anti-Fn1, magenta) counterstaining on *Tg(kdrl:GFP)* and Elastinb (anti-Elnb, magenta) counterstaining on *Tg(fli:lifeact-EGFP)* showing their expressions in the OFT at 72 hpf (**B**), (**B'**) respectively) at 96 hpf (**C**), (**C'**) respectively) and at 120 hpf (D, D' respectively). Scale bar: 20 µm. Arrows show

*Figure 3 continued on next page*

*Figure 3 continued*

the Fn1 localisation within the valve leaflets. Results obtained from three independent experiments. (E) Scheme of the three layers shown in A', B, C and D (magenta, smooth muscles; green, endothelium; grey, myocardium; Fibronectin1, magenta lines) at 72hpf, 96hpf and 120hpf.

DOI: https://doi.org/10.7554/eLife.44706.005

The following figure supplement is available for figure 3:

**Figure supplement 1.** Smooth muscle identity is revealed by NO.

DOI: https://doi.org/10.7554/eLife.44706.006

We next investigated the expression of the Hippo effector Yap1 and a reporter of the Hippo pathway in vivo. We found that Yap1 is localized in the heart at 72 hpf, in particular in the OFT smooth muscle cells (*Figure 5—figure supplement 1A,B*). To better characterize its tissue-specific expression, we used the double transgenic line *Tg(fli:lifeact-EGFP; kdrl:nls-mCherry)* and could show that the smooth muscle cells surrounding the OFT, but also some OFT endothelial cells express Yap1 (asterisks in *Figure 5—figure supplement 1B*). To assess the activity of the Hippo pathway in the heart at 72 hpf, we made use of the *Tg(4xGTIIC:d2GFP)* and could see that the Yap/Wwtr1-Tead reporter was activated in the OFT endothelial cells (labelled using the *kdrl:membrane-mCherry* line) as well as the smooth muscle cells surrounding the OFT (labelled by the Elnb staining) (*Figure 5A*). To assess the implication of Yap1 during OFT valve development, we followed the same embryos over time and looked at the valve phenotype in *yap1* mutant embryos, *yap* embryos and *yap1* control embryos (*Agarwala et al., 2015*) (*Figure 5B*). When analyzed from 72 hpf until 120 hpf, a significant fraction of *yap1-/-* embryos displayed abnormal valves (17% at 72 hpf, 8% at 96 hpf, 17% at 120 hpf, n = 12) and an increasing fraction of *yap1* embryos did not have recognizable OFT valves (58% at 120hpf, n = 12). Thus, these results suggest that *yap1* is involved during OFT valve morphogenesis (*Figure 5B*) and that smooth muscle cell progenitors are likely to play a role in the process.

Together, these results show that Klf2a, Notch signaling, and Hippo pathways are active in the OFT during valve morphogenesis.

## Klf2a and notch reporter activity is flow-dependent in the OFT

As blood flow is an important regulator of *klf2a* expression and cardiac valve formation, we next wanted to assess whether changes in flow properties impact Klf2a and the Notch reporter activity in the OFT.

We analyzed the reporters activity following injection of a morpholino specific for *troponin T2a* (*tnnt2a*), which is necessary for heart contraction and reliably mimics the *sih* mutants (*Sehnert et al., 2002*), to determine whether these signaling pathways were impacted when heart contraction is abnormal and/or are activated upon shear stress forces. As the absence of heart contraction can impact heart morphogenesis, we injected highly diluted *tnnt2a* morpholino (hypomorphic condition) into these two reporter lines (*Figure 6A,B*). Such treatment allows us to decrease heart function and flow without dramatically altering heart shape (*Figure 6A,B*, *Videos 1* and *2*). Depending on the knockdown efficiency in single embryos, this treatment leads to the generation of two groups of embryos: group1 where the heart beats 'normally' (normal heart rate at 2–3 Hz and function) and group2 where the heartbeat is slower. In the group of 'beating heart' embryos (group1, *Video 1*), we still observe stronger GFP expression in the posterior part of the valve for both reporters at every stage analyzed (*Figure 6A*). In the group of embryos where the heart is still beating but at an abnormal slow rate (less than 2 Hz, group2, *Video 2*), we observe no difference between the anterior and posterior part of the valve (p=0.99) for the Klf2a reporter at 72 hpf (*Figure 6B*). For the Notch reporter, we observed no difference in fluorescence intensity at 72 hpf (p=0.1) (*Figure 6B*). In addition, we analyzed the localization of Fn1, Elnb and Yap1 by immunostaining in both 'beating heart' (n = 4, n = 7 and n = 7, respectively) and 'slow beating heart' (n = 6, n = 7 and n = 7 respectively) groups and could observe that Fn1, Elnb, and Yap1 are down-regulated in the smooth muscle cells of the 'slow beating heart' embryos (*Figure 6D,F*). Moreover, we assessed the BA diameter and the activity of the smooth muscle using the DAF-FMDA assay (*Figure 6E*). The results suggest that the 'slow beating heart' group has a smaller BA (p<0,001) and the smooth muscle are much less active (p=0,05) (*Figure 6E,F*). We next assessed valve morphology in which blood flow is altered due to slow heart contraction and selected the fish with almost no contraction (group2). All the *tnnt2a*MO-

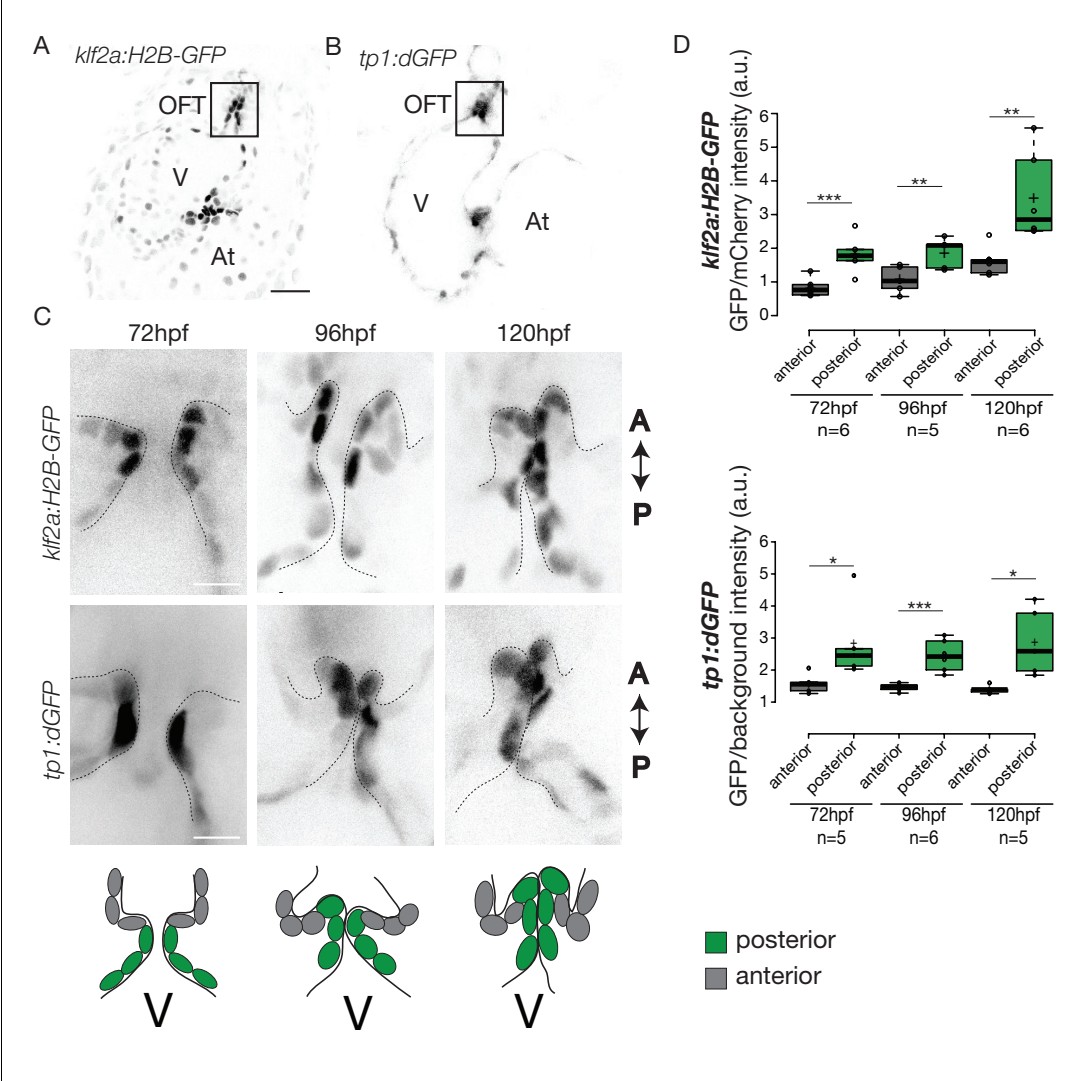

**Figure 4.** Klf2a and Notch reporters are activated in the OFT endothelium Confocal z-section of the *Tg(klf2a:H2B-GFP; fli:nls-mCherry)*. (**A**) and *Tg(tp1: dGFP)* (**B**) at 72hpf. OFT: outflow tract, At: atrium, V: ventricle. Scale bar: 20 µm. (**C**) Confocal z-section of OFT valves expressing the Klf2a reporter and Notch reporter at 72 hpf, 96 hpf, and 120 hpf. Schemes explaining the considered anterior (grey) and posterior (green) parts of the valve at 72 hpf, 96 hpf, and 120 hpf. A: anterior, P: posterior. V: ventricle. Scale bar: 10 µm. (**D**) Quantification of the fluorescent intensity of the Klf2a (GFP over mCherry) and Notch (GFP over background) reporters in the anterior versus posterior part of the valves at 72 hpf (n = 6 embryos, p=0001 and n = 5 embryos, p=0,02 respectively), 96 hpf (n = 5 embryos, p=0005 and n = 6 embryos, p=0,0007 respectively) and 120 hpf (n = 6 embryos, p=0,01 and n = 5 embryos, p=0,01) in wild-type embryos using the student's t-test. Boxplots: Center lines show the medians; box limits indicate the 25th and 75th percentiles as determined by R software; whiskers extend 1.5 times the interquartile range from the 25th and 75th percentiles, outliers are represented by dots. Results obtained from three independent experiments.

DOI: https://doi.org/10.7554/eLife.44706.007

The following source data and figure supplement are available for figure 4:

**Source data 1.** Fluorescence intensity measurements.

DOI: https://doi.org/10.7554/eLife.44706.009

**Figure supplement 1.** Expression of the *klf2a* reporter Z-section of the *Tg(klf2a:H2B-EGFP; kdrl:nls-mCherry)* at 72 hpf, 96 hpf and 120 hpf used to quantify the reporter expression.

DOI: https://doi.org/10.7554/eLife.44706.008

injected embryos have no valve (n = 9/9, $p<10^{-6}$) (*Figure 6C*). To confirm the role of flow in the process, we analyzed the effect of altered blood viscosity and shear stress by lowering red blood cell content in the *gata1* mutants (*Vlad Tepes*) as previously described (*Steed et al., 2016b*;

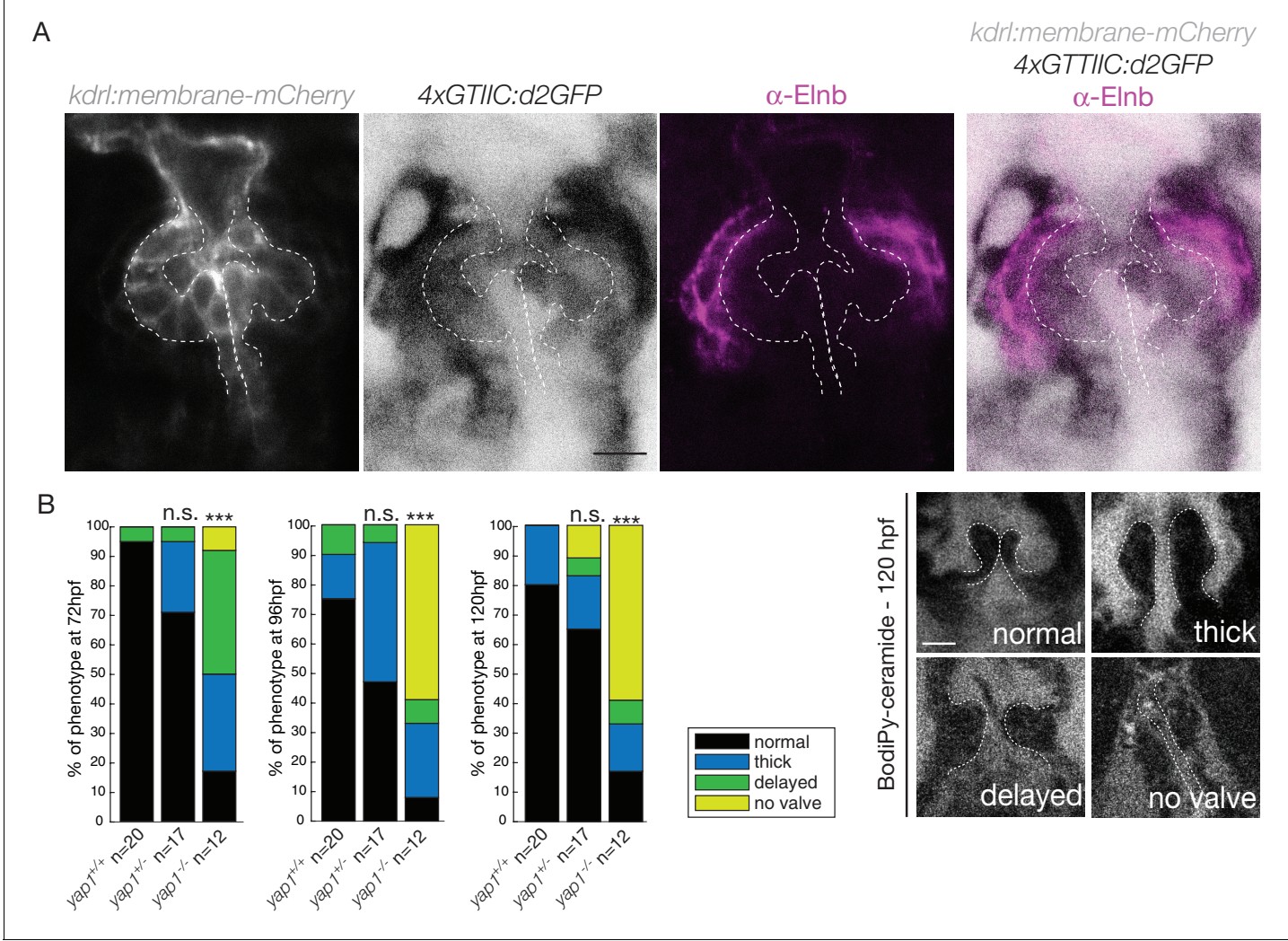

**Figure 5.** Hippo pathway effector Yap1 is active and Yap1 is essential for valve formation in the OFT. (**A**) Confocal z-sections of the double transgenic line *4xGTIIC:d2GFP; kdrl:membrane-mCherry* counterstained with the Elnb antibody and focused on the OFT. Scale bar: 20 µm. (**B**) Example of the valve phenotypes (normal, thick, delayed and no valve) and quantification of the phenotypes in the *yap1^{+/+}* controls embryos, *yap1^{+/-}* and in *yap1^{-/-}* mutant embryos. Chi-square test. n.s.: non significant, \*\*\*: $p < 10^{-3}$. Scale bar: 10 µm. Results obtained from two independent experiments.

DOI: https://doi.org/10.7554/eLife.44706.010

The following source data and figure supplement are available for figure 5:

**Source data 1.** Phenotypic quantifications.
DOI: https://doi.org/10.7554/eLife.44706.012
**Figure supplement 1.** Yap1 is expressed in the OFT Yap1 antibody staining on *Tg(fli:lifeact-EGFP; kdrl:nls-mCherry)* (**A**) and zoom on the OFT (**B**) at 72 hpf.
DOI: https://doi.org/10.7554/eLife.44706.011

*Vermot et al., 2009*). In *vlad tepes* mutant embryos (n = 21, $p < 10^{-2}$), more than 80% of the mutants displayed abnormal OFT valves (*Figure 6C*).

Together, these results suggest that the expression of both Klf2a and Notch reporters is flow-dependent and that mechanical forces associated with heart activity are necessary for valve development and smooth muscle cell identity in the OFT.

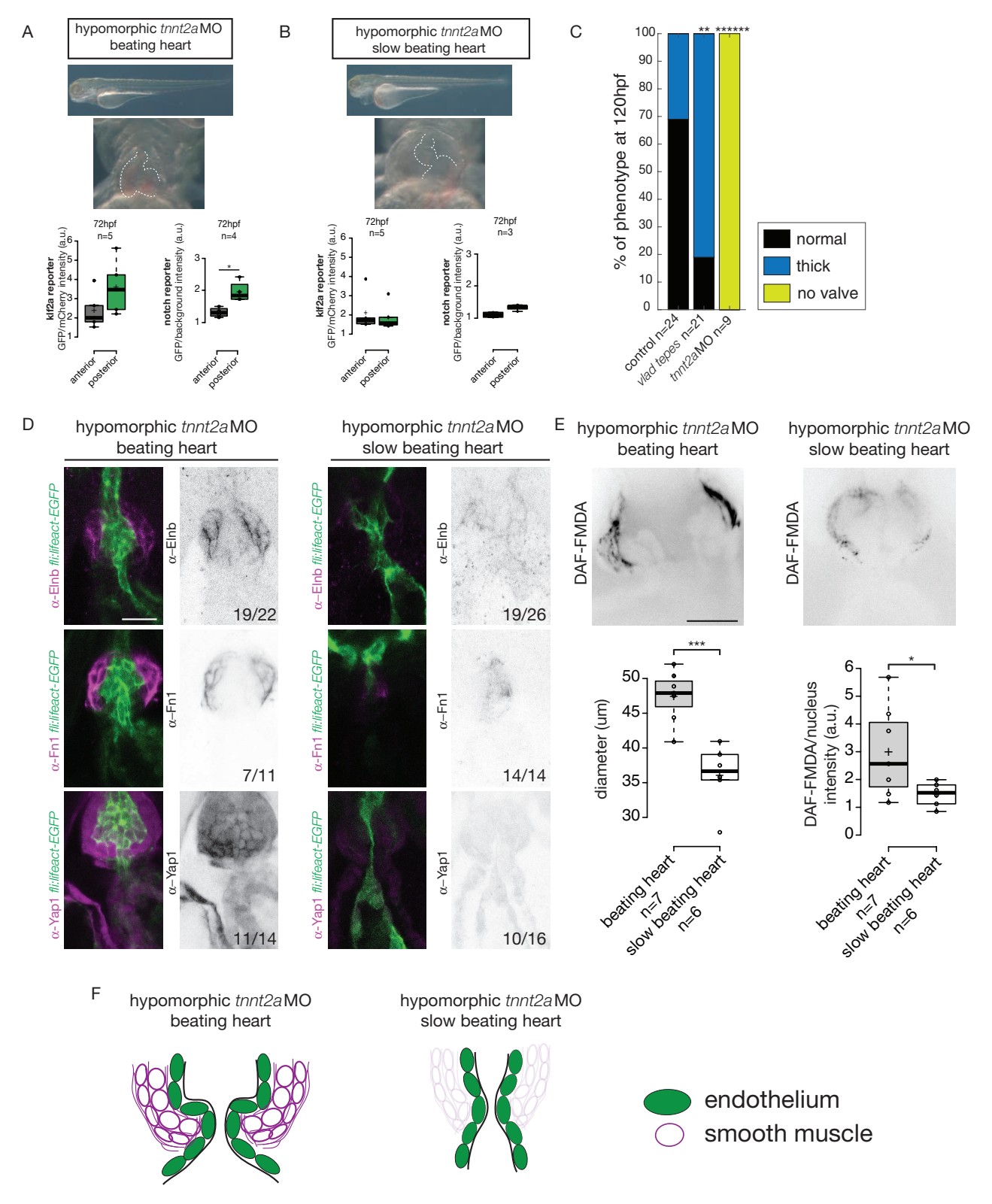

**Figure 6.** Klf2a and notch response, as well as the smooth muscle cell identity, are flow-dependent. Quantification of the Klf2a and Notch reporter expressions in *tnnt2a*-morpholino injected embryos showing a 'beating heart' (p=0,27 and p=0,01 respectively). (**A**) and a 'slow beating heart' (p=0,7 and p=0,1 respectively) (**B**) at 72 hpf. N = 2 independent experiments. (**C**) Quantification of the phenotypes in the control (n = 24), vlad tepes mutant (n = 21 embryos from two independent experiments), *tnnt2a*MO-injected embryos (n = 9 embryos from two independent experiments). Chi-square test.

*Figure 6 continued on next page*

*Figure 6 continued*

**: $p<10^{-2}$, ******: $p<10^{-6}$. (D) Z-sections of the *Tg(fli:lifeact-eGFP)* counterstained with either Fibronectin1 (Fn1), elastin (Elnb) or Yap1 in *tnnt2a*-morpholino injected embryos (slow beating and beating heart). Scale bar: 20 µm. N = 2 independent experiments. (E) Z-section and quantification of the BA diameter and DAF-FMDA intensity in tnnt2a-morpholino injected embryos (p=0,0005 and p=0,05 respectively). Student's t-test. Boxplots: Center lines show the medians; box limits indicate the 25th and 75th percentiles as determined by R software; whiskers extend 1.5 times the interquartile range from the 25th and 75th percentiles, outliers are represented by dots. (F) Scheme summarizing the down-regulation of the smooth muscle markers in 'slow beating heart' embryos compared to 'beating heart" embryos.
DOI: https://doi.org/10.7554/eLife.44706.013

The following source data is available for figure 6:

**Source data 1.** Phenotypic and fluorescence reporters quantifications.
DOI: https://doi.org/10.7554/eLife.44706.014

## Klf2a regulates OFT valve morphogenesis via notch signaling activation and is necessary for the smooth muscle cells differentiation

As *notch1b* and *klf2a* are expressed in the OFT endothelium in response to flow forces, we hypothesized that they have a role during OFT valve morphogenesis. Therefore, we imaged the *Tg(fli:gal4/UAS:kaede)* and *Tg(flli:lifeact-EGFP; kdrl:nls-mCherry)* transgenic lines (*Figure 7A,B*). We looked at the phenotype of OFT valve endothelium in the *klf2a⁻/⁻* and *notch1b⁻/⁻* embryos at 72 hpf, 96 hpf and 120 hpf (*Figure 7A,B*). We found that most of the *klf2a⁻/⁻* embryos display proper valves at 72 hpf (n = 7/12), 96 hpf (n = 6/12) and 120 hpf (n = 6/12). However, 33% of the *klf2a⁻/⁻* embryos have abnormal 'delayed phenotype' at 72 hpf (n = 4/12), at 96 hpf (n = 4/12) and 120 hpf (n = 4/12) (*Figure 7A*). In this case, the OFT valves are larger, leading to big cushions instead of thin valve leaflets (*Figure 7A*). In addition, *notch1b* is also necessary for proper valve formation since 30% (n = 6/20) of the *notch1b⁻/⁻* embryos have a 'delayed phenotype' at 72 hpf, 35% (n = 7/20) at 96 hpf and 45% (n = 9/20) at 120 hpf (*Figure 7B*) while almost all control embryos have proper valves at 72 hpf (n = 6/7), 96 hpf (n = 7/7) and 120 hpf (n = 7/7) (*Figure 7B*). Next, we wondered whether a cross-regulation between *notch1b* and *klf2a* exists since they are both necessary for proper valve formation. First, we compared *notch1b* expression between *klf2a⁻/⁻* embryos and *klf2a⁺/⁺* control embryos (*Figure 7—figure supplement 1A*). In controls, 85% and 91% of the embryos have *notch1b* expression in the OFT at 48 hpf and 72 hpf, respectively (*Figure 7—figure supplement 1A–C*). However, in *klf2a⁻/⁻* embryos, *notch1b* expression is altered with no clear expression defining the OFT region at 48 hpf (64%) and at 72 hpf (78%) (*Figure 7—figure supplement 1A,B*). However, in the reverse experiment, most of the embryos show proper *klf2a* expression in *notch1b -/-* embryos (61% at 48 hpf and 51% at 72 hpf) (*Figure 7—figure supplement 1C*).

In order to assess the regulation of the Notch pathway activity by Klf2a in the endothelium, we analyzed the Notch reporter activity in the posterior and anterior parts of the valve in *klf2a⁺/⁺* (n = 5) versus *klf2a⁻/⁻* (n = 4) embryos. As for Notch reporter expression in wild-type (*Figure 4C,D*), the Notch reporter is significantly more expressed in the posterior part compared to the anterior part of the valves at 72 hpf ($p<10^{-2}$), 96 hpf ($p<10^{-2}$) and 120 hpf ($p<10^{-1}$) (*Figure 7C,C'*) in the *klf2a⁺/⁺* embryos. Interestingly, the stronger posterior expression is lost in the *klf2a⁻/⁻* embryos and the fluorescent intensity in the posterior part of the valve is significantly reduced in the *klf2a⁻/⁻* compared to *klf2a⁺/⁺* at 72 hpf (p<10–1), 96 hpf ($p<10^{-1}$) and 120 hpf ($p<10^{-1}$) (*Figure 7C,C'*). In comparison, the expression in the posterior part of the valve in the *notch1b⁻/⁻* embryos compared to the *notch1b⁺/⁺* control embryos is not significantly different at any time point analyzed (*Figure 7—figure supplement 1D*). To further assess whether Klf2 has an effect on the OFT formation, we performed Fn1, Elnb and Yap1 stainings on *klf2a⁻/⁻* and controls at 72

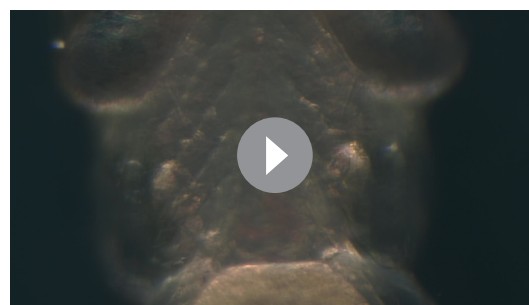

**Video 1.** Bright field videomicroscopy of the typical heart function of a *tnnt2a* MO group1 embryo where the heart is still beating normally.
DOI: https://doi.org/10.7554/eLife.44706.015

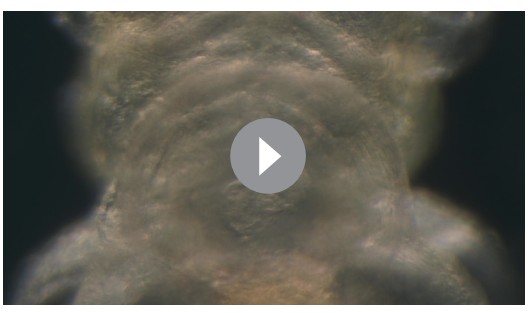

**Video 2.** Bright field videomicroscopy of the typical heart function of a *tnnt2a* MO group2 embryo where the heart is still beating but at an abnormal slow rate (less than 2 Hz).
DOI: https://doi.org/10.7554/eLife.44706.016

hpf. We could observe that Fn1, Elnb, and Yap1 are properly localized in controls (n = 13/13, n = 7/7 and n = 4/4, respectively), *klf2a*[-/-] (n = 5/6, n = 3/4, and n = 3/3) (*Figure 7—figure supplement 1E*).

These results suggest that *klf2a* and *notch1b* are involved for proper valve morphogenesis and that Klf2a modulates *notch* expression in the process. Moreover, *klf2a* does not seem necessary for the smooth muscle cell differentiation surrounding the endothelium.

## Piezo channels regulate both the endothelial and smooth muscle cell markers expression

In order to decipher whether flow and mechanosensitive channels could be involved in the regulation the OFT valve formation, we proceeded to analyze the phenotype of different potent mechanosensitive channel mutants. We focused on the non-selective ion channels Trpp2, Trpv4, Piezo1 and Piezo2a (*Figure 8—figure supplement 1A,B,C*) that are known for their mechanosensitive properties (*Coste et al., 2010*; *Köttgen et al., 2008*; *Li et al., 2014*; *Sharif-Naeini et al., 2009*; *Thodeti et al., 2009*). First, we evaluated the relative fractional shortening (RFS) at 72hpf in the atrium and in the ventricle of each mutant and their corresponding controls (*Figure 8—figure supplement 1A*). We did not observe any significant difference in the RFS between mutants and their respective controls neither in the atrium nor in the ventricle, suggesting that heart function in these mutants is normal. As another readout of flow forces and heart function, we quantified the retrograde flow fraction (RFF) at 72hpf and 120hpf in these mutants and the time windows for forward, reverse or no flow. We found that they are equivalent in all mutants when compared to their respective controls (*Figure 8—figure supplement 1B*), confirming that heart function is not different in these mutants. We looked at valve morphology at 120 hpf when leaflets are normally fully formed. In control embryos, the valves are extending into the lumen (*Figure 8—figure supplement 1C*). We found that 33% of the *trpv4* mutant embryos (n = 7) have normal valves and mainly display thick valves (67%). In the *trpp2* mutant embryos (n = 20), a stronger phenotype is observed with only 10% of normal valves. We observed that some embryos are delayed with respect to valve phenotype, meaning that the valve forming area still displays cushions at 120 hpf instead of having leaflets (35%). The *trpv4*[-/-]; *trpp2*[+/-] (n = 8) has an intermediate phenotype with 25% of the embryos having normal valves. Finally, the *trpp2*[-/-]; *trpv4*[-/-] embryos (n = 11) have a stronger phenotype, with none of the embryos showing proper OFT valve development and displaying mainly a delayed valve phenotype (45%). Interestingly, the *piezo1* mutant embryos (n = 14) show mainly a delayed valve formation (50%), similarly to the *trpv4*[-/-]; *trpp2*[-/-] embryos. The *piezo2a*[-/-] (n = 9) has a less stringent phenotype with mainly normal valves (44%) but nevertheless 11% of *piezo2a*[-/-] fish do not have valves at all. This phenotype is even more prevalent in *piezo1*[-/-]; *piezo2a*[-/-] embryos (n = 11), where none of the fish display proper valve development and most of them do not form any valves (36%).

To better characterize the valve phenotype in *trpp2*[-/-] and *piezo1*[-/-] embryos, we made use of the *Tg(fli:lifeact-EGFP; kdrl:nls-mCherry)* transgenic line and assessed the shape of the endothelium at 72 hpf (*Figure 8A* and *Figure 8—figure supplement 2A*), 96 hpf (*Figure 8B* and *Figure 8—figure supplement 2A*) and 120 hpf (*Figure 8C* and *Figure 8—figure supplement 2A*). *trpp2*[-/-] embryos display mainly thick valves at all time points (n = 4/8 at 72 hpf, n = 8/13 at 96 hpf and n = 5/11 at 120 hpf) (*Figure 8A–C* and *Figure 8—figure supplement 2A*). Indeed, the endothelial layer is not a single cell layer anymore (*Figure 8B*) which leads to a thicker leaflet at 120 hpf (*Figure 8C*). Although *piezo1*[-/-] embryos have mainly normal valves at 72 hpf (*Figure 8A* and *Figure 8—figure supplement 2A*) (n = 6/10), they display a delayed phenotype (still cushions) at 96 hpf (*Figure 8B* and *Figure 8—figure supplement 2*) (n = 4/10) and thick or delayed phenotype at 120 hpf (*Figure 8C* and *Figure 8—figure supplement 2A*) (n = 3/10 and n = 3/10 respectively). To assess the redundancy between *trpp2* and *piezo1*, we performed the same experiment in *piezo1*[-/-]; *trpp2*-morpholino injected embryos at 72 hpf (*Figure 8A*), 96hpf (*Figure 8B*) and 120 hpf (*Figure 8C*). Although the

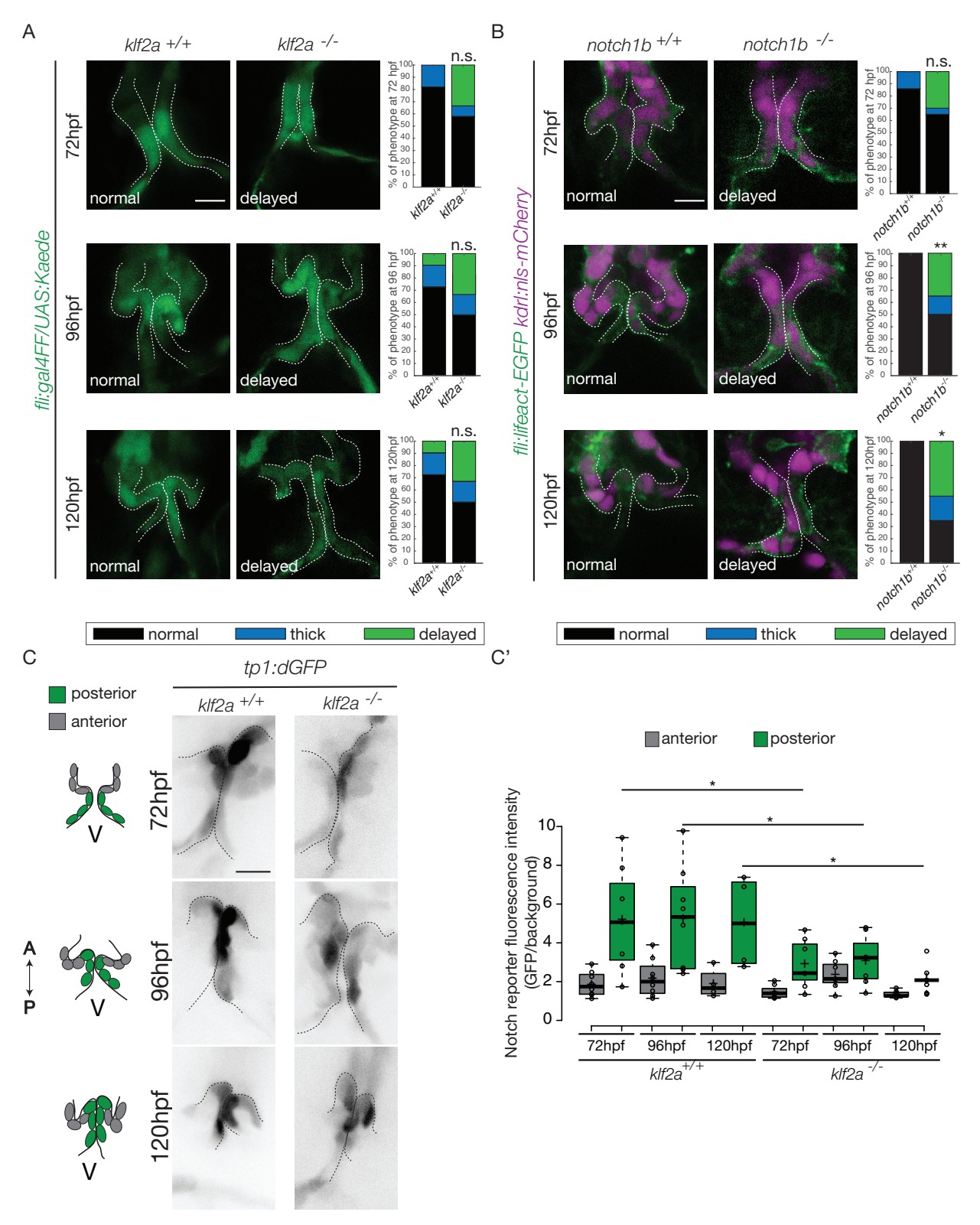

**Figure 7.** Klf2a and notch are necessary for valve formation. Quantification of the valve phenotypes at 72 hpf, 96 hpf and 120 hpf (normal, thick, delayed) in *klf2a*+/+(n = 11), and *klf2a*-/- (n = 12) using the *Tg(fli:gal4FF/UAS:Kaede).* (A) and *notch1b*+/+ (n = 7) and *notch1b*-/- (n = 20) using *Tg(fli: lifeact-EGFP; kdrl:nls-mCherry)* embryos. (B) Scale bar: 10 μm. N = 3 independent experiments. (C) Confocal z-sections of the *Tg(tp1:dGFP)* in *klf2a*+/+ and *klf2a*-/- embryos at 72 hpf, 96 hpf and 120 hpf. V: ventricle. Scale bar: 10 μm. (C') Quantification of the fluorescent intensity of the Notch reporter

*Figure 7 continued on next page*

*Figure 7 continued*

(GFP over background) in the anterior versus posterior parts of the valves in in *klf2a*$^{+/+}$ (n = 5) and *klf2a*$^{-/-}$ (n = 4) embryos. Statistical test were performed to compare the posterior intensities in *klf2a*$^{+/+}$ versus *klf2a*$^{-/-}$ at 72 hpf (p=0,05), 96 hpf (p=0,03) and 120 hpf (p=0,04). Student's t-test. Boxplot: Center lines show the medians; box limits indicate the 25$^{th}$ and 75$^{th}$ percentiles as determined by R software; whiskers extend 1.5 times the interquartile range from the 25$^{th}$ and 75$^{th}$ percentiles, outliers are represented by dots. Results obtained from three independent experiments.
DOI: https://doi.org/10.7554/eLife.44706.017

The following source data and figure supplement are available for figure 7:

**Source data 1.** Fluorescent intensity quantifications.
DOI: https://doi.org/10.7554/eLife.44706.019
**Figure supplement 1.** Klf2a regulates *notch1b* expression but Notch1b does not regulate *klf2a* expression.
DOI: https://doi.org/10.7554/eLife.44706.018

*trpp2*-MOrpholino injected embryos do not show a phenotype as strong as the *trpp2*$^{-/-}$ mutant embryos at 96 hpf and 120 hpf (possibly due to less effective morpholinos at these stages), the double *piezo1*$^{-/-}$; *trpp2*-MO do not clearly show a stronger phenotype than the single *trpp2*$^{-/-}$ or *piezo1*$^{-/-}$ embryos (normal valves in n = 6/7, n = 2/7 n = 4/7 and at 72 hpf, 96 hpf and 120 hpf respectively). These results suggest that Trp and Piezo channels are necessary for the proper folding of the endothelium from 72 hpf to 120 hpf.

To better characterize the cell layer affected by *piezo1* loss of function, we performed immunohistochemistry against Yap1 and two smooth muscle identity markers (Fn1 and Elnb) as well as their active functionality in *piezo1*$^{+/+}$ controls and *piezo1*$^{-/-}$ mutants at 72 hpf (**Figure 8D** and **Figure 8— figure supplement 2B**). A bit more than two thirds of *piezo1*$^{-/-}$ embryos (n = 7/10) display reduced Yap1 expression as well as a down-regulation of Elnb (n = 9/12) and Fn1 (n = 3/4) (**Figure 8E**). However, the BA diameter as well as the DAF-FMDA fluorescence intensity are not affected in the *piezo1*$^{-/-}$ compared to *piezo1*$^{+/+}$ (**Figure 8D** and **Figure 8—figure supplement 2B**) demonstrating that smooth muscle cells are still present in the piezo mutants. These results suggest a selective role of *piezo1* in the regulation of the smooth muscle cell maturation and proper Yap1 localization in the OFT. We next assessed Klf2a reporter expression in *piezo1*$^{-/-}$ and *trpp2*$^{-/-}$. We found that Klf2a expression was misregulated in *piezo1*$^{-/-}$ mutant embryos with stronger Klf2a reporter activation in the anterior and posterior part of the valve endothelium when compared to controls (p<10$^{-3}$ for the anterior part and p<10$^{-2}$ for the posterior part of the valves) (**Figure 8F,F'**). By contrast, we found that the posterior part of the valve has a decreased expression of GFP (p<10$^{-2}$), highlighting a down-regulation of *klf2a* in the *trpp2*$^{-/-}$ embryos (**Figure 8G,G'**). These results suggest that Piezo1 inhibits *klf2a* overall the valve endothelium (**Figure 8F,F'**) while Trpp2 is required for *klf2a* activation in the posterior part of the OFT valve endothelium (**Figure 8G,G'**). To assess if the localization of *piezo1* and *trpp2* mRNA could explain the differential function of these channels, we performed RNAscope assay at 72hpf (**Figure 8—figure supplement 3A**). We found that *trpp2* is ubiquitously expressed in the embryo, including in the different layers composing the OFT. Similarly, we found that *piezo1* is expressed in both endothelium and smooth muscles, albeit at a lower level than *trpp2* (**Figure 8—figure supplement 3A**). To confirm these results, we generated a transgenic reporter line with 3 kb of the *piezo1* promoter upstream of the start codon (*piezo1:nls-Venus*). We observed the expression of the reporter line mostly in smooth muscles at 72hpf (**Figure 8—figure supplement 3B**) and cells of the endothelium of the OFT valve. Trpp2 immunohistochemistry showed that Trpp2 is expressed in the endothelium and the smooth muscles confirming that *trpp2* is ubiquitously expressed in the OFT (arrow in **Figure 8—figure supplement 3B**). We conclude that Piezo1 plays a dual role in the OFT: it modulates *klf2a* expression in the endothelium and it is necessary for proper smooth muscle cell maturation around the OFT endothelium.

## Discussion

Using cardiac live imaging and functional studies combined with in vivo reporter analysis, we uncover key mechanosensitive signaling pathways involved in OFT valve morphogenesis (**Figure 9**). We identify two tissue layers sensitive to mechanical forces in the OFT: (1) The endothelial cells where *klf2a* expression is modulated both by Piezo and Trp channels (2) The smooth muscles surrounding the endothelium, where Piezo channels regulate Yap1 localization and smooth muscle cell specific

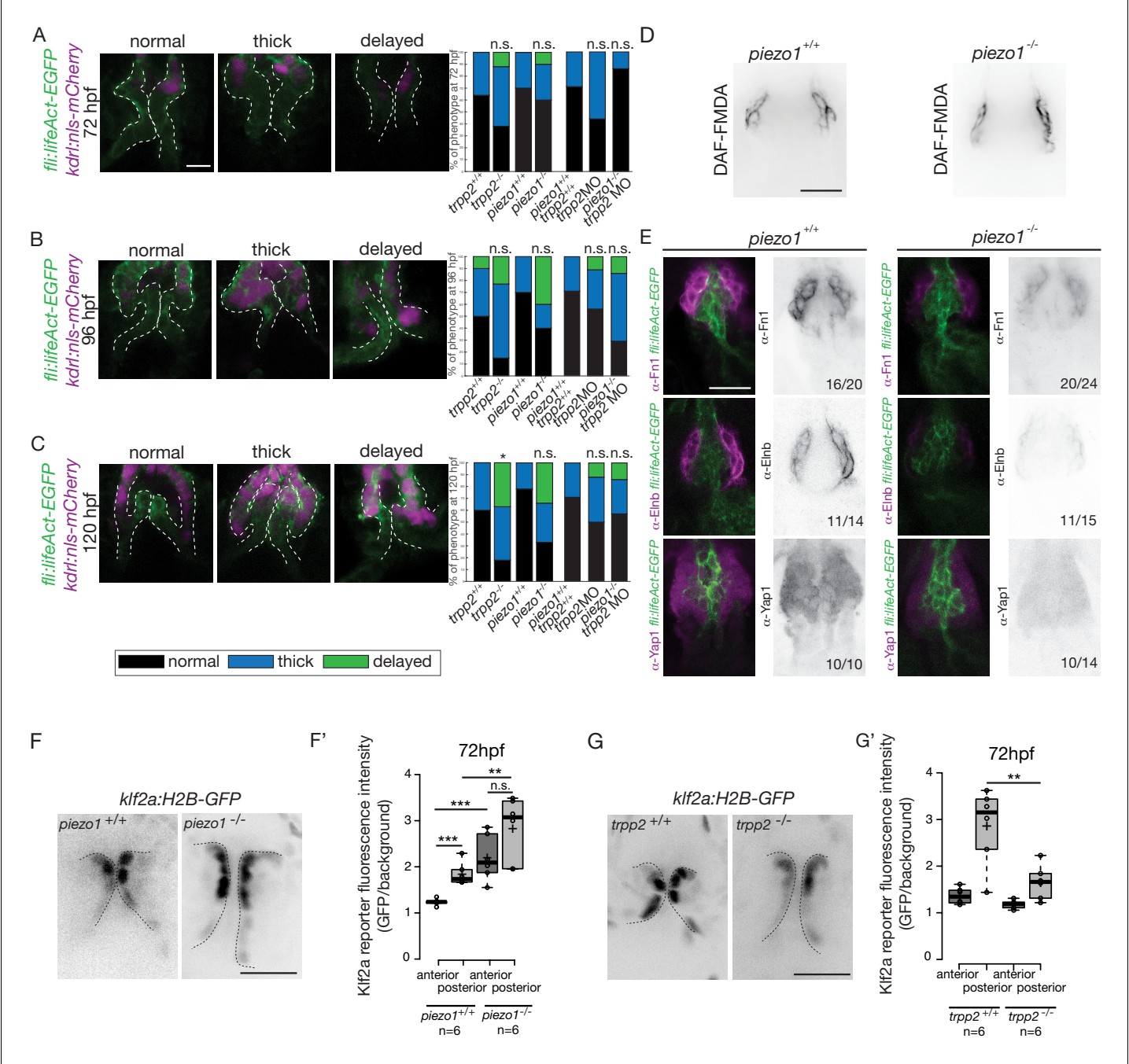

**Figure 8.** Flow and mechanosensitive channels are necessary for proper OFT valve formation. (**A**) Z-sections and quantifications of the valves phenotypes (normal, thick, delayed) of the *Tg(fli:lifeaAct-EGFP; kdrl:nls-mCherry)* at 72 hpf (**B**), 96 hpf (**C**) and 120 hpf (**D**) in *trpp2*⁺/⁺(n = 11, n = 10, n = 10), *trpp2*⁻/⁻ (n = 8, n = 13, n = 13 from three independent experiments), *piezo1*⁺/⁺ (n = 10, n = 10, n = 9), *piezo1*⁻/⁻(n = 10, n = 10, n = 9 from two independent experiments), *piezo1*⁺/⁺; *trpp2*⁺/⁺ (n = 7), trpp2-morpholino injected embryos (n = 9) and *piezo1*⁻/⁻; *trpp2*-morpholino injected embryos (n = 7). Scale bar: 10 µm. (**D**) Z-section of the OFT stained with DAF-FMDA in *piezo1*⁺/⁺ and *piezo1*⁻/⁻. Scale bar: 20 µm. (**E**) Fibronectin1 (Fn1), elastin (Elnb) and Yap1 staining on *Tg(fli:lifeact-eGFP)* in *piezo1*⁺/⁺ (n = 12, n = 4 and n = 10 respectively) and *piezo1*⁻/⁻ (n = 12, n = 4 and n = 10 respectively from three independent experiments). Scale bar: 20 µm. Z-sections (**F**) and quantification (**F'**) of the klf2a reporter (GFP over background) in the anterior and posterior parts of the valves in *piezo1*⁺/⁺ (n = 6) and *piezo1*⁻/⁻ (n = 6) obtained from two independent experiments. Scale bar: 20 µm. Z-sections (**G**) and quantification (**G'**) of the *klf2a* reporter (GFP over background) in the anterior and posterior parts of the valves in *trpp2*⁺/⁺ (n = 6) and *trpp2*⁻/⁻ (n = 6) (obtained from two independent experiments). Scale bar: 20 µm. Student's t-test. Boxplot: Center lines show the medians; box limits indicate the 25th and 75th percentiles as determined by R software; whiskers extend 1.5 times the interquartile range from the 25th and 75th percentiles, outliers are represented by dots.

*Figure 8 continued on next page*

*Figure 8 continued*

DOI: https://doi.org/10.7554/eLife.44706.020

The following source data and figure supplements are available for figure 8:

**Source data 1.** Phenotypic and fluorescence reporters quantifications.

DOI: https://doi.org/10.7554/eLife.44706.024

**Figure supplement 1.** Embryo phenotypes in controls and mutants.

DOI: https://doi.org/10.7554/eLife.44706.021

**Figure supplement 2.** Valve phenotypes in trpp2$^{-/-}$ and in piezo1$^{-/-}$.

DOI: https://doi.org/10.7554/eLife.44706.022

**Figure supplement 3.** Trpp2 and Piezo1 expression in the OFT.

DOI: https://doi.org/10.7554/eLife.44706.023

marker expression (*Figure 9*). These observations enable us to confirm the universal role of mechanical forces in cardiac valve morphogenesis and suggest a specific mechanism for OFT valve morphogenesis in which the origins of the valve progenitors, the implication of particular groups of cells, the mechanosensors involved and the impact of the mechanotransduction cascade are identified.

## Klf2a modulates notch signaling specifically in the OFT endothelium

Valve morphogenesis occurs in complex mechanical environments. In the AVC, endocardial cells experience both shearing forces and mechanical deformation due to the contraction of the heart and its associated blood flow. The situation is slightly different in the OFT because endothelial valvular progenitors are not surrounded by contractile cardiomyocytes but passive smooth muscle cells

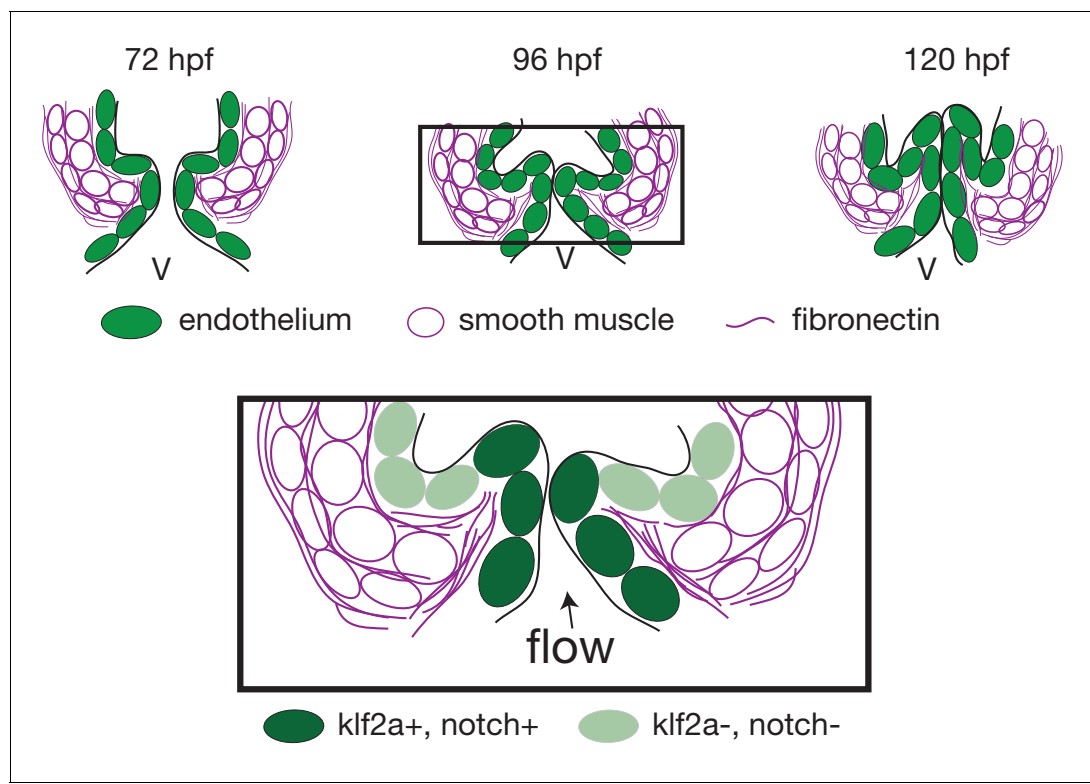

**Figure 9.** Working model summarizing OFT valve morphogenesis in response to mechanical forces. Two cell layers forming the OFT respond to piezo1 activity: the endothelium (green) and smooth muscle cells (magenta). Endothelial *klf2a* expression is repressed by Piezo1. In the smooth muscle cells, the expression of elastin (Elnb), fibronectin (Fn1) and Yap1 is modulated by Piezo1. Fibronectin is localized in the smooth muscle cell layer and within the forming valve (magenta lines). Future work will help to decipher how the two cell layers interact with each other to modulate OFT valve formation. V, ventricle.

DOI: https://doi.org/10.7554/eLife.44706.025

that can provide a counter force to flow and pressure. Here, we show that the main mechanosensitive pathways involved in AVC valve development are required for OFT valve development, even though the mechanical stimuli vary greatly between these two sets of valves. Previous studies have proposed that *klf2a* and *notch1b* are important during the formation of functional cardiac valve leaflets in zebrafish. Both are specifically expressed in the AVC (*Beis et al., 2005*; *Pestel et al., 2016*; *Steed et al., 2016b*; *Vermot et al., 2009*) and are transcriptionally misregulated in models of Cerebral cavernous malformation (CCM) where cardiac valve development is altered (*Donat et al., 2018*). Here, we show that both Notch signaling and *klf2a* are active in the endothelial cells of the OFT between 48 hpf and 120 hpf. During the process in which zebrafish cardiac AVC cushion remodel into valve leaflets, endocardial Klf2a expression and Notch activity are high on the luminal side of the developing valve leaflet, which is exposed to blood-flow, whereas their expression is lower on the abluminal side of the leaflet (*Pestel et al., 2016*; *Steed et al., 2016b*). In the OFT, Klf2a expression and Notch activation follow a different pattern because we could not clearly identify abluminal cells in the OFT. This might reflect a complete lack of endothelial to mesenchymal transition in the OFT by comparison to the AVC. Nevertheless, the impact of both pathways on valve morphogenesis remains the same as we found that *klf2a*$^{-/-}$ and *notch1b*$^{-/-}$ embryos show similar valve phenotypes and Notch signaling pathway is significantly decreased in *klf2a*$^{-/-}$ mutants. Even though the downstream players of the notch pathway remain to be established in the OFT, these results validate previous observations, suggesting that *klf2a* acts upstream of *Notch* signaling in the endocardium (*Donat et al., 2018*; *Samsa et al., 2015*; *Vermot et al., 2009*). Considering that Notch has been proposed to be mechanosensitive in blood vessels (*Mack et al., 2017*), an attractive hypothesis is that the Klf2-Notch axis could be a general mechanosensitive cascade in endothelial cells. In that case, it would be interesting to address *notch* activity in a flow related context, similarly to what is currently being done with Nf-kb, Klf2a, and other flow responsive pathways (*Feaver et al., 2013*). This assumption is particularly interesting in the context of sprouting angiogenesis and other aspects of angiogenesis where Notch is broadly required (*Choi et al., 2017*; *Hasan et al., 2017*; *Pitulescu et al., 2017*; *Tammela et al., 2011*). We further show that the expression of Fibronectin1 and Elastinb in the smooth muscle cells surrounding the OFT is not altered in the *klf2a*$^{-/-}$embryos. Considering that Piezo is emerging as an important contributor to human diseases related to blood cells (*Ma et al., 2018*; *Zarychanski et al., 2012*) and lymphatic diseases in humans (*Lukacs et al., 2015*), our work should motivate the search of potential involvement of Piezo in cardiac pathologies such as valvulopathies.

## Piezo is necessary for proper OFT smooth muscle cell identity and endothelial *klf2a* expression in the OFT

Cardiac valve development is highly dependent on endothelial/endocardial cell mechanosentivity. Previous work identified the membrane-bound mechanosensitive channels (Trpp2 and Trpv4) and a calcium-activated intracellular signaling cascade leading to Klf2a expression and valve morphogenesis as key elements of the endocardial mechanodetection-signaling pathway (*Heckel et al., 2015*) Consistently, we found that these Trp channels are also required for endothelial Klf2a expression and OFT valve formation. In addition, we identify another type of mechanosensitive channel belonging to the Piezo family. Piezo channel mutants show valve dysgenesis phenotype, suggesting their requirement during OFT valve morphogenesis. Interestingly, Piezo channels are important both for modulating Klf2a expression in the endothelium and for smooth muscle cell-specific expression of Elnb and Fn1. Piezo function in these two cell layers is consistent with the fact that Piezo1 is expressed in smooth muscle cells and is important for tissue remodeling in mouse arteries (*Retailleau et al., 2015*) as well as in endothelial cells in mouse vasculature (*Li et al., 2014*; *Ranade et al., 2014*) and lymphatic valves (*Nonomura et al., 2018*). These studies along with our results suggest that Piezo1 might be necessary for both layers to regulate distinguishable functions: activation of signaling cascade upon shear stress in the OFT endothelium and proper cell identity acquisition in smooth muscles. Both tissue layers might be sensing different stimuli: shear plus strain for the endothelial cells and strain for the smooth muscle cells. This hypothesis is consistent with the fact that Piezo is sensitive to stretch (*Ranade et al., 2015*), compression (*Lee et al., 2014*; *Qi et al., 2015*), and rhythmic mechanical stimuli (*Lewis et al., 2017*). Importantly, Piezo-dependent mechanisms can transduce forces at the cell-cell or cell-matrix interface (*Eisenhoffer et al., 2012*; *Poole et al., 2014*). Thus, Piezo can have different mechanosensitive roles both in smooth muscle

cells and endothelium to coordinate the expression of Klf2 and ECM proteins. An interesting hypothesis is that the smooth muscle cell layer participates in the shaping up of the OFT valve and that both tissue layers establish paracrine interactions to fine tune the morphogenetic process. Further work will be needed to identify if this is the case and how it affects OFT valve morphogenesis at the cellular scale.

## Yap1 and hippo pathways are regulators of the OFT valve formation in the endothelium and smooth muscles

Our work shows that Yap1 is specifically localized in the OFT in both endothelial cells and smooth muscle cells and is required for valvulogenesis. Accordingly, the Hippo pathway is active in smooth muscles and in the endothelium. Our work suggests that Piezo constitutes a plausible regulator of Yap1 localization as Piezo1 seems important for its localization in smooth muscle cells. Interestingly, Yap1 has been shown to translocate less in the nucleus in Piezo1 mutant mice neural stem cells (*Pathak et al., 2014*) and to induce proliferation in smooth muscles during cardiovascular development in mouse (Wang et al., 2014). Even though we were not able to assess Yap1 subcellular localization in vivo, the regulation of Yap1 activity and ECM assembly might be a general feature of Piezo function. The connection between Piezo and Yap1 is particularly interesting in the context of OFT development and function. In teleost, the OFT has an important biomechanical role for the control of flow propagation within the vascular networks by contributing to the dampening of the pressure wave down the arterial tree (*Braun et al., 2003b*). In zebrafish, Yap1 is involved in the determination of cardiac precursor cells into smooth muscle cell fate via a process that involves the regulation of Elastinb expression (*Moriyama et al., 2016*). Besides the control of cell identity, ECM contributes to biomechanical properties of tissues (*Dzamba and DeSimone, 2018*). It is thus possible that Piezo acts as a regulator of tissue mechanical properties by regulating Yap1 expression both in the OFT and in the vascular system where Yap1 expression is flow inducible (*Nakajima et al., 2017*).

In summary, this study reveals a novel function for mechanosensitive Piezo and Trp channels in modulating OFT valve development as well OFT smooth muscle cell maturation. It will be important to further investigate endothelial-smooth muscle cells interactions during OFT cardiac wall maturation.

# Materials and methods

## Zebrafish strains, husbandry, embryo treatments, and morpholinos

Animal experiments were approved by the Animal Experimentation Committee of the Institutional Review Board of the IGBMC (reference numbers MIN APAFIS#4669–2016032411093030 v4 and MIN 4669–2016032411093030 v4-detail of entry 1). Zebrafish lines used in this study were *Tg(fli1a:lifeact-EGFP)* (*Phng et al., 2013*), *Tg(kdrl:nls-mCherry)* (*Nicenboim et al., 2015*), *Tg(kdrl:EGFP)* (*Jin et al., 2005*), *Tg(fli1a:nls-mCherry)* (*Heckel et al., 2015*), *Tg(myl7:egfp)* (*Huang et al., 2003*), Tg(−26.5Hsa. WT1-gata2:eGFP)$^{cn12}$ (*Sánchez-Iranzo et al., 2018*), *Tg(klf2a(6 kb):H2B-eGFP)* (*Heckel et al., 2015*), *Tg(tp1:dGFP)* (*Ninov et al., 2012*), *Tg(4xGTIIC:d2GFP)* (*Miesfeld and Link, 2014*), *vlad tepes$^{m651}$* (*Lyons et al., 2002*), *cup$^{tc321}$* (*Schottenfeld et al., 2007*), *piezo1$^{sa12608}$* (EZRC), *piezo2a$^{sa12414}$* (ZIRC), *trpv4$^{sa1671}$* (ZIRC), *notch1b$^{sa11236}$* (ZIRC), *klf2a$^{ig4}$* (*Steed et al., 2016b*), *yap1$^{fu48}$* (*Agarwala et al., 2015*) and wild-type AB. Cup mutant embryos were phenotyped based on the curled tail phenotype. The *Tg(piezo1:nls-Venus)* was generated by injection of the *piezo1:nls-Venus* plasmid and the mRNA of the Tol2 transposase. The plasmid was generated by cloning of 3 kb of the zebrafish *piezo1* promoter upstream of the ATG start site and 3xnls-Venus into a pTol2-GAGGS vector. All animals were incubated at 28.5°C for 5 hr before treatment with 1-phenyl- 2-thiourea (PTU) (Sigma Aldrich) to prevent pigment formation. Morpholino specific for *tnnt2a* (*Sehnert et al., 2002*) (5'-CATGTTTGCTCTGATCTGACACGCA-3') were obtained from GeneTools. It was injected into the yolk at the one-cell stage at a concentration of 5,8 ng to stop the heart. It was diluted 40 times in order to get fish with either a decreased heartbeat ('slow beating heart' group in this study) or close to the non-injected fish heartbeat ('beating heart' group in this study).

## Immunofluorescence

Embryos were fixed at the desired stage in 4% paraformaldehyde overnight at 4°C. After washing in 1X PBST (PBS-0.1% Tween-20), embryos were permeabilized in 1X PBST containing 1% Triton-X 100 for 30 min at room temperature or overnight at 4°C. For Fibronectin1 staining, the pericardial cavity was then carefully pierced with the tip of a needle to facilitate antibody penetration before blocking in permeabilization buffer supplemented with 5% BSA. Embryos were incubated in blocking solution containing 5% BSA and 15% NGS (α-Fn1), 1% BSA, 2% NGS (anti-Elnb) and 2% BSA, 2% MgCl2 (1M), 5% NGS supplemented by 1,5% Tween-20 (anti-Yap1) for 2 hr at room temperature or overnight at 4°C. Primary antibodies were added to the relevant blocking solution and incubated 2 overnights at 4°C. Secondary antibodies were added in blocking solution after thorough washing in PBST and incubated for 2 days at 4°C. Embryos were thoroughly washed in PBST and mounted for imaging on a Leica SP8 confocal. Antibodies were used as follows: rabbit α-Fn1 (F3648, Sigma) 1:100, rabbit anti-Elnb (*Miao et al., 2007*; kind gift from Burns lab; *Paffett-Lugassy et al., 2017*) 1:1000, rabbit anti-Yap1 (generated by the Lecaudey lab; *Agarwala et al., 2015*) 1:300, rabbit anti-Trpp2 (kind gift from Drummond lab) 1:100 and Alexa Fluor 647 goat anti-rabbit IgG secondary antibody (A21245, Life Technologies) were used at 1:500.

## In situ hybridization

In situ hybridization was performed as in *Thisse and Thisse (2008)*. Anti-sense probes for *notch1b* and *klf2a* were generated from a plasmid containing the cDNA of zebrafish *notch1b* in pCR-script SK+ (provided by the Bakkers lab, The Netherlands) and zebrafish *klf2a* in IRBOp991B0734D (provided by RPDZ, Berlin; *Vermot et al., 2009*) and subsequently transcribed using the T3 polymerase and T7 polymerase, respectively. After ISH, embryos were incubated subsequently in 45% and 90% D-fructose (Sigma F0127) containing 0.5% of 1-Thioglycerol (Sigma M6145) for 20 min. Imaging of ISH was done using a Leica M165 macroscope with a TrueChrome Metrics (Tucsen) with a Leica 1.0X objective (10450028).

## RNAscope

72 hpf wildtype zebrafish embryos were fixed in 4% PFA overnight at 4°C. The fixed embryos were dehydrated to 100% ethanol gradually. Embryos were stained using the RNAscope Fluorescent Multiplex kit (Advanced Cell Diagnostics).

## Confocal imaging

For live imaging, zebrafish embryos were staged, anaesthetized with 0.02% tricaine solution or 50 mM BDM, to stop the heart when necessary, and mounted in 0.7% low melting-point agarose (Sigma Aldrich). Confocal imaging was performed either on a Leica SP8 confocal microscope (experiments with BODIPY-ceramide or fixed samples) or a Leica spinning disk (valve structure, flow profile, reporter experiments). Fast confocal imaging to image valve leaflets from 72 hpf to 120 hpf stained with BODIPY-ceramide was performed using the resonant scanner mode of the Sp8 microscope. Images were acquired with a low-magnification water immersion objective (Leica HCX IRAPO L, 25X, N.A. 0.95). The optical plane was moved 2 μm between the z-sections until the whole OFT was acquired. 2-colored fast confocal imaging was used to image valve structure, red blood cells, and reporter activities from 56 hpf to 144 hpf was performed using a Leica DMi8 combined with a CSU-X1 (Yokogawa) spinning at 10 000 rpm, two simultaneous cameras (TuCam Flash4.0, Hamamatsu) and a water immersion objective (Leica 20X, N.A. 0.75 or Leica 40X, N.A. 1.1). 1 ms exposure was used for red blood cells imaging and 20 ms exposure for valve structure and reporter activity experiments. 50% of 488 laser power and 40% of 561 laser power were used for reporter activity experiments.

## BODIPY-ceramide imaging

Embryos were incubated with 4 mM BODIPY-ceramide (Molecular Probes) overnight and then processed as in *Heckel et al. (2015)* and *Vermot et al. (2009)* to visualize the valve shape.

## DAF-FMDA labelling

To reveal the presence of NO, embryos were incubated in zebrafish medium containing 5 µM DAF-FM DA (Life Technologies, D23842) for 30 min in the dark at 28°C. Fluorescence intensities of the smooth muscles were measured using ImageJ software.

## Photoconversion

Photoconversion was performed using the FRAP module on a SP8 confocal microscope and a Leica HCX IRAPO L, 25X, NA0.95 water immersion objective. *Tg(fli1a:Gal4FF; UAS:Kaede)* embryos were mounted in 0.7% low melting-point agarose supplemented with 50 mM BDM to inhibit heart contraction for the duration of the procedure. A region of interest corresponding to the anterior, middle or posterior part of the valve was selected and exposed to 405 nm light (25% laser power). One pre-bleach frame was acquired, followed by 3–6 bleach pulses (3–5 ms each) without acquisition to achieve conversion of the kaede protein to its red form. A z-stack of the photoconverted heart was then acquired in the standard confocal mode to record the starting point of each experiment. Embryos were then carefully dissected from the agarose, placed in fish water for 5–10 min until heart contraction resumed and then put at 28.5°C to develop individually under standard conditions until the next time point of interest.

## Fractional shortening

Imaging was performed on a Leica DMIRBE inverted microscope using a Photron SA3 high-speed CMOS camera (Photron, San Diego, CA) and water immersion objective (Leica 20X, NA 0.7). Image sequences were acquired at a frame rate of 1000 frames per second. FS% = (Dm diastole- Dm systole)/ (Dm diastole) and where Dm is the diameter of the chamber of interest (atrium or ventricle).

## Flow analysis

Red blood cells were manually tracked through the OFT and their velocity calculated from image sequences of the *Tg(gata1:ds-red; kdrl:EGFP)* beating heart, acquired at 1000 frames per second as described previously. Red blood cells transiting through the OFT were tracked manually on Imaris and their velocity calculated. The tracks of multiple cells in at least four embryos per stage were assembled to obtain an estimate of the flow velocity over multiple cardiac cycles (typically 3). Velocities estimated at the same time point by tracking different cells were averaged.

## Image analysis

For fluorescence intensity analysis of the Klf2a reporter, the *Tg(klf2a:H2B-EGFP)* reporter line was crossed with the *Tg(fli1a:nls-mCherry)* line and the mCherry signal was used for normalization. The maximum intensity of each channel on a single Z-section through the valves was quantified and the *EGFP over mCherry ratio generated. For fluorescence intensity analysis of the Notch reporter, the ratio of the maximum intensity of the GFP signal from the *Tg(tp1:dGFP)* reporter line on a single Z-section through the valves over the maximum intensity of the background was generated.

These ratios were then averaged for three cells in the anterior part and three cells in the posterior part of both valves in the OFT of individual embryos. Finally, the averages of the anterior and posterior parts were compared.

## Statistical analyses

We did not compute or predict the number of samples necessary for statistical differences because the standard deviation of our study's population was not known before starting our analysis. Biological replicate corresponds to the analysis of different embryos of the same stage. Technical replicate corresponds to the analysis of the same embryo imaged the same way. The sample size (biological replicate and number) to use was as defined by our ability to generate our datasets. For analyses between two groups of embryos, differences were considered statistically significant when the p-value<0.05, as determined using a two-tailed and paired Student's t-test (*klf2a* and *notch* reporter expression). For boxplots, center lines show the medians, crosses show the means, box limits indicate the 25th and 75th percentiles as determined by R software; whiskers extend 1.5 times the interquartile range from the 25th and 75th percentiles, data points are represented as circles.

## Acknowledgements

We thank the Vermot laboratory for discussion and H Fukui and R Chow for thoughtful comments on the manuscript. We thank C Burns, V Lecaudey and I Drummond for providing antibodies, mutants and protocols for immunohistochemistry. We thank J-M Garnier for the cloning of the *piezo1:nls-Venus* construct. We thank the IGBMC fish facility (S Pajot and C Moebs) and the IGBMC imaging center, in particular B Gurchenkov, D Hentsch, E Guiot and E Grandgirard. This project has received funding from the European Research Council (ERC) under the European Union's Horizon 2020 research and innovation programme: GA N°682939, the Fondation pour la Recherche Médicale: DEQ29553, Agence Nationale de la Recherche: ANR-15-CE13-0015-01, ANR-10-IDEX-0002–02, ANR-12-ISV2-0001-01 and ANR-10-LABX-0030-INRT and the European Molecular Biology Organization Young Investigator Program. HV was supported by the IGBMC International PhD program: ANR-10-LABX-0030-INRT. ALD was supported by a post doctoral fellowship from the Lefoulon-Delalande Foundation.

## Additional information

### Funding

| Funder | Grant reference number | Author |
| --- | --- | --- |
| H2020 European Research Council | 682938 - EVALVE | Julien Vermot |
| Fondation pour la Recherche Médicale | DEQ29553 | Julien Vermot |
| Agence Nationale de la Recherche | ANR-15-CE13-0015-01 | Anne-Laure Duchemin<br>Hélène Vignes<br>Julien Vermot |
| European Molecular Biology Organization | Young Investigator Program | Julien Vermot |
| Fondation Lefoulon Delalande | | Anne-Laure Duchemin |
| Agence Nationale de la Recherche | ANR-10-IDEX-0002-02 | Anne-Laure Duchemin<br>Hélène Vignes<br>Julien Vermot |
| Agence Nationale de la Recherche | ANR-12-ISV2-0001-01 | Anne-Laure Duchemin<br>Hélène Vignes<br>Julien Vermot |
| Agence Nationale de la Recherche | ANR-10-LABX-0030-INRT | Anne-Laure Duchemin<br>Hélène Vignes<br>Julien Vermot |

The funders had no role in study design, data collection and interpretation, or the decision to submit the work for publication.

### Author contributions

Anne-Laure Duchemin, Conceptualization, Data curation, Formal analysis, Methodology, Writing—original draft; Hélène Vignes, Data curation; Julien Vermot, Conceptualization, Supervision, Funding acquisition, Writing—original draft, Project administration

### Author ORCIDs

Julien Vermot https://orcid.org/0000-0002-8924-732X

### Ethics

Animal experimentation: Animal experiments were approved by the Animal Experimentation Committee of the Institutional Review Board of the IGBMC.(reference numbers MIN APAFIS#4669-2016032411093030 v4 and MIN 4669-2016032411093030 v4-detail of entry 1).

Decision letter and Author response
Decision letter https://doi.org/10.7554/eLife.44706.028
Author response https://doi.org/10.7554/eLife.44706.029

## Additional files

### Supplementary files
• Transparent reporting form
DOI: https://doi.org/10.7554/eLife.44706.026

### Data availability
Source data for the figures has been uploaded.

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
