## [Decision Letter]

Thank you for sending your article entitled "Mechanically activated Piezo channels control outflow tract valve development through Yap1 and Klf2-Notch signaling axis" for peer review at *eLife*. Your article is being evaluated by three peer reviewers, one of whom is a member of our Board of Reviewing Editors, and the evaluation is being overseen by Didier Stainier as the Senior Editor.

Given the list of essential revisions, including new experiments, the editors and reviewers invite you to respond within the next two weeks with an action plan and timetable for the completion of the additional work. We plan to share your responses with the reviewers and then issue a binding recommendation.

Whilst all reviewers find merit in your work, two of the reviewers unanimously raise concerns regarding the low number of replicates in many experiments, both in those that use signalling reporters for klf2 and notch that are deemed inherently highly variable, and those that use mutants to assess phenotypes. Please see the details below.

Furthermore, the reviewers are concerned that the overall model is not sufficiently supported by the data, and that the phenotypic descriptions appear somewhat overstated. There is also a perceived disconnect between the described effects on the smooth muscle cells and the endothelial cells, with insufficient insight into how the mechanosensitive ion channels regulate the phenotypes of either of the two, and no information of potential collective effects or interaction in the morphogenic process.

The reviewers find that based on these weaknesses, the overall level of mechanistic insight is unsatisfactory.

The reviewers and the BRE have discussed extensively whether the necessary revisions will be feasible within the normal revision period. There is a consensus in that there could be benefits in delineating more carefully either the klf2a Notch axis or the mechanisms of the ion channels to provide focus and the level of detail required to substantiate the model.

For more information on the details, we have appended the full comments of the reviewers below.

Reviewer #1:

The manuscript by Duchemin and colleagues analyses the morphogenic events of outflow tract development during embryogenesis in the zebrafish. The authors are experts in this area and have a strong track record in studying flow dependent signalling and morphogenic regulation in the vasculature and the heart in fish. They exploit their excellent imaging skills to describe the changing shape of the outflow tract, tracking endothelial cells by photoconversion and imaging between 72 and 144hpf. The authors report that the endothelium shows a prominent tissue folding process without signs of delamination, likely driven by morphogenic events between both the endothelium and the surrounding smooth muscle cells, but unlikely involving the myocardium. Matrix labelling highlights prominent expression of fibronectin by the smooth muscle cells during the process.

The authors use klf2a and tp1 reporter lines to address when and where Klf2a and Notch signalling is activated, identifying a strong upregulation in the posterior part of the valve formation. Slowing heart function with hypomorphic Tnnt2a MO demonstrates that both signals are flow dependent, whilst mutants identify that klf2a is important for Notch activity, but not vice versa. The authors conclude that Klf2a regulates valve morphogenesis via Notch and show that Notch activity loses some of its polarity in klf2a mutants. Using Yap and tead reporter fish, the authors further show that hippo signalling is active in the smooth muscle cells and also some endothelial cells. Here the resolution is less clear in the Yap1-reporter, but the data do suggest that the smooth muscle cells show the strongest signal. Yap signal is also flow dependent, and YAP mutants like Klf2a mutants show delayed or defective valve formation at varying frequency. Searching for upstream activators of these mechanosensitive pathways in endothelial cells and smooth muscle, the authors finally investigate mutants for 4 mechanosensitive ion channels, Piezo1, Piezo2a, Trpp2 and TrpV4.

The results suggest some level of redundancy or cooperativity, with Piezo1 and Trpp2 important for endothelial folding, and Piezo1 involved in smooth muscle cell activation of Yap1 to modulate outflow tract development.

The work is overall very carefully executed, includes all relevant controls, beautifully illustrated and very comprehensively presented throughout. The details of endothelial morphogenesis, gene regulation, and the mechanosensitive signalling cascade provide for a much deeper understanding of this developmental process. The schematic illustrations accompanying the original imaging data is extremely helpful. Overall, this is an outstanding piece of work that is deemed of interest for not only the cardiovascular development community, but also for those interested in translational aspects given the importance of inherited cardiac defects associated with various mutations in pathways addressed in this work.

If I was to point out a weakness of the work, it would be the lack of commenting and possibly analysing the crosstalk between the smooth muscle cells and the endothelium in the process. The final concluding schematic places Piezo1 upstream of different components in both cell compartments, but given that the downstream effectors of Yap1 include modulators of matrix, Bmp and tgfb signalling etc., which will impact on mechanical properties of both endothelium and the smooth muscle cells, and that Notch activity also regulates similar targets, it would seem justified to comment on potential paracrine interactions between the two. I realize however that a full investigation here would go beyond the scope of the current work.

A second aspect that deserves a comment would be how Piezo1 inhibits klf2a whilst Trpp2 is deemed a positive regulator. How is this dichotomy achieved?

Finally, the kdrl:nls-mcherry line appears to label only very few endothelial cells in the posterior OFT and the images here do not really match the schematic. I assume the authors use the fli:lifeact GFP signal to delineate those kdrl negative cells, but this would seem important to comment for technical reasons. On a mechanistic level, it would be important to comment on this cellular heterogeneity in the posterior OFT?

Reviewer #2:

In the submitted manuscript, Duchemin and colleagues tackle the formation of the OFT valve in the zebrafish model and its dependency on mechanic stimuli. Their work indicates a first rudimentary pathway of mechanosensation-implicated players in the formation of the OFT valve, and describes the different morphogenesis of the OFT valve compared to the classically studied AV canal valve in the zebrafish heart.

Overall, the work raises general questions about the overall mechanism of flow-dependent morphogenetic processes. The study covers several valuable points, including the first analysis of piezo mutants in zebrafish. The figures include data points and schematics that will be valuable for future discussions and work in the field.

Nonetheless, at times, the manuscript makes strong conclusions based on small sample sizes (i.e. total embryos analyzed, see below) and the expression of isolated marker genes. Several mutant phenotypes have highly variable penetrance and expressivity (i.e. yap1 mutants, subsection “Klf2, Notch signaling, and Hippo pathways are active in the OFT in different cell layers and are all necessary for proper OFT valve development”), the analysis of which is again hampered by low sample sizes and ambiguous description as to how mutants are identified (assumed Mendelian ratio? Genotyping of individual embryos? etc.). Several statements of necessity and "dramatic" phenotypes need reassessment, as the phenotypes rather indicate the studied genes contribute to robust OFT valve formation rather than being necessary (i.e. *notch1b, klf2*; see also below). Further, OFT valve formation is a somewhat specialized developmental process, yet the authors seem tempted to draw broad-stroke generalizations about mechanosensing from their findings in the Discussion.

* The authors analyze several mutants and state necessity and importance for valve formation. Overall, however, the phenotype analysis is plagued by several issues as presented, and rather suggests the studied factors might be at times dispensable yet contribute to robust valve formation.

In all figures with box diagrams, at times vast variability is captured (i.e. Figure 8G'), yet individual data points not shown. Individual data points should be added. Further, the overall sample size in the majority of the mutant analyses seem low and spotty in the reported details, is shown in percent that are misleading in small sample sizes, and affects the phenotype interpretation. Percentage is used with exceedingly small sample sizes and becomes misleading in significance. Similarly for p-values, that are not sufficient to cover vastly incomplete penetrance of phenotypes. As examples:

a) The yap1 mutants have a seemingly increasing penetrance and expressivity of OFT valve phenotypes – yet the phenotype is reported for only 12 embryos (also, are these always the same 12 embryos?), resulting in 58% (6 or 7) of embryos with no valves at 120 hpf; that hardly seems to justify an "important role" but rather a contribution to robust valve formation.

b) Mutant *notch1b* embryos have a 30% incidence for valve issues, also not warranting the moniker "necessary" but rather contributing. Similarly, the authors state "most" *notch1b* embryos show proper klf2 expression, yet it's about half at only n=4 (Figure 7).

c) The authors state that a "large fraction" of piezo1 embryos have less Yap1 at n=7/10, again seeming like an overstatement (can also be called "a bit more than half"?).

Such examples go throughout all mutant analysis (also trpp2, etc.) and should be re-phrased and possibly remedied by increasing the sample size to achieve better insights into variability of the phenotypes.

Also, how do yap1, piezo, trpp2, etc. mutants look like overall? A description of other phenotypes (i.e. cardiac edema, endothelial problems, viability, etc.) would greatly help to gauge the non-autonomous impact of the described defects (or their dependence thereof).

* The authors conclude that loss of piezo function impacts smooth muscle "identity" of the BA based on downregulated Fn1 and Elnb staining. While interesting, Fn1 and Elnb are functional/differentiation markers and not necessarily determinants of fate/identity (i.e. Moriyama et al., 2016), yet these observations might rather indicate that flow has a functional contribution to BA maturation. The authors are encouraged to perform a functional BA assay, i.e. active NO metabolism using DAF-2DA (Grimes et al., 2006) to underline the functional impact on BA formation. Further, the claim of Fn1 expression in smooth muscle is not well-funded based on the provided images in Figure 3 and should be shown more clearly.

Reviewer #3:

This manuscript addresses how mechanical forces and genetic programs interact to control valve morphogenesis. The authors assess in vivo morphogenesis of the OFT valve and how mechanosensitive pathways: Yap, Klf2, and Notch1b contribute to this process. Finally they test the role of mechanosensors from the Trp and Piezo family.

While the topic of the manuscript is very interesting and timely, the conclusions drawn from the data are not based on enough evidence.

Throughout, the number of independent experiments should be clearly stated; the n in most of the experiments is very low. This makes any rigorous assessment futile and greatly undermines the quality of the manuscript.

All the quantification/bar graphs should show individual data points wherever possible. The statistical tests should be clearly stated in the figure legends. The manuscript text would benefit from revision; e.g. the last sentence of the Introduction is not clear, or adjectives as reversing flow or tissular remodeling are used and are not correct English.

1) In the Introduction: "The multichambered heart contains two different set of valves: arterial valves that are semilunar and mitral (atrioventricular) valves that are tricuspid." Where is the reference to that? Mitral valve is also called the left atrioventricular valve. Tricuspid valve is also called the right atrioventricular valve. Only mitral valve is bicuspid, all other valves are tricuspid in 4 chambered heart. In zebrafish all valves are bicuspid, they are at IFT, AVJ, and OFT. The authors do not mention IFT valve for some reason at all. The valve anatomy in 4 vs 2 chambered heart should be corrected and properly referenced; a simple schematic would be helpful for clarity.

2) "… the developmental programs driving mitral valve…,less is known about arterial valves" The reference Wu et al. cites more than 20 genes involved in arterial valve development. The rest of the Introduction mentions several signaling pathways involved in aortic valve formation. So meanwhile it is not entirely true that less is known about arterial valves. The Introduction would benefit if it is shortened and refocused on the current knowledge of the effect of mechanical forces on valvulogenesis.

3) Figure 1A-C: in the text a different transgenic is referred to then shown in the panels. What is the time point in Figure 1A?

4) Figure 2B: in the third panel, there are photoconverted cells anteriorly and posteriorly, where is the anterior cell located at 120 hpf? The photoconverted cells should be highlighted by a star in all panels, not just in one.

Consecutive panels for at least 3 embryos should be shown in the supplement.

5) The claim that AVC valve is delaminating should be referenced. Scherz et al., 2008, showed that in zebrafish AVC valve is forming by invagination and not via endocardial cushion formation, similarly to the process of OFT formation that the authors observe here. The conclusion that the OFT valve morphogenesis is unique is thus overstated. It is not clear what the authors mean when they say "unique".

6) The rationale that invagination of the OFT endothelium might be aided by the adjacent tissue should be introduced more clearly. In Figure 1 and 2 only endothelial contribution is observed and shown, yet the authors claim that the cellular contribution of the valve lead them to the hypothesis of adjacent tissue contribution. What is the model?

In Figure 3, besides merged images, individual channels for Fibronectin, Elastin b, and endothelial marker should be shown. Counterstaining for Fibronectin and Elastin b should be performed together; from the images presented, not all fibronectin positive cells seem to be Elastin b positive, and the schematic is therefore misleading. If panel A' (Figure 3) is a zoom of panel A, and both are scale bar of 10 microns, why is the scale bar in panel A longer than in panel A'? it does not look like much of a zoom either. This should be revised.

The vascular smooth muscle cells of BA of the OFT are accrued to the ventricle after 48 hpf. This process was recently described e.g. by Felker et al., 2018. Even though the OFT valve formation is occurring concomitantly with the formation of BA, these two events should not be equated, as stated in the conclusions describing Figure 3. If the authors wished to compare OFT valve formation to AVC valve formation, they should provide additional panels of AVC valve for comparison or at least briefly recap the events of AVC valve formation.

7) What is the stage in Figure 4A and B? The reporters should be visualized with the markers for endothelium and elastin b. The channel for fli:nls-mCherry should be shown especially if the GFP/mCherry ratio is used for the quantification in 4D. How is anterior/posterior boundary defined at different stages for the purpose of the quantification in 4D if the endothelium is invaginating in the direction from posterior to anterior as the authors show in Figure 2? This seems to be arbitrary and misleading due to the dynamic nature of the forming valve. The division medial/lateral or luminal might be more appropriate. What are the individual data points in D? the n should be increased for 120hpf for klf2 reporter; n=2 is not enough! The description of how exactly the quantification was performed should be included, as this read-out is used in the subsequent experiments; is the intensity for both leaflets per embryo averaged? From a single section or z-stack?

Statistical test should be named in the figure legend.

8) In Figure 5, the GTIIC:d2GFP Yap reporter should be used together with the endothelial and nuclear marker and elastin b; from the images presented it is not clear how the dashed line was drawn in 5B. Furthermore, Yap reporter should be used together with the nuclear marker, and nuclear/cytoplasmic ratio should be quantified. The antibody staining for Yap appears to be cytoplasmic, which does not confer anything about its signaling activity.

The description of 5C should be rewritten; "…a significant fraction of yap1 mutants displayed abnormal valves (17% at 72 hpf…", – 17% show normal valves, the text is misleading. The quantification of the phenotypes should be performed from at least 3 independent experiments, so that appropriate statistical test can be performed.

Yap1 mutants should be properly referenced.

If the Yap reporter is active in both endothelial cells and smooth muscle cells of BA, the statement that smooth muscle progenitors are likely to play a role in valve morphogenesis is a speculation and should be corrected. That Yap is essential for OFT valve development is an overstatement when 40% of the mutant embryos still form some sort of a valve structure.

Is the 5A' zoom of A? the ROI in A does not correspond to A'?

9) The myl7:GFP reporter line should be used to determine the heart morphology upon different tnnt2 MO doses. From the images presented nothing can be concluded about the MO effect, except perhaps a mild edema in B. The MO doses should be indicated either in the figure directly or figure legend. The BA size should be measured under these MO conditions; it is known that cardiac function may affect the BA size. What is the cut off for the heart rate in the "slow-beating" group? Also the authors should be consistent with the naming, use group 1/2 or beating/slow beating, but not both. The sample number of klf2 and notch reporter has to be increased; the reporters, especially the ones using destabilized GFP are very variable, n of 3 with the usual clutches of at least 100 zebrafish embryos under normal breeding conditions is just not acceptable. The activity of klf2 and notch reporter should be assessed at later time point – 120 hpf when the flow effects are more pronounced, assuming that the data presented are at 72 hpf; this information is missing.

In the experiments assessing the Fb1, Eln B and Yap abundance, both the smooth muscle cells of BA and endothelial cells have to counterstained, e.g. NO fluorescent indicator DAR4-AM is a reliable marker of the BA smooth muscle cells. It is not clear if the smooth muscles cells are present at all with the high dose of tnnt2 MO, and as such the "missing" staining can be simply a result of missing cells. In Figure 6D, at what time point the valve presence was quantified? n=9 for tnnt2 MO is very low; are the authors really injecting just 9 embryos?

The controls are completely missing in the experiments presented in Figure 6, with the exception of Figure 6D.

While the conclusion that klf2 and notch activity are flow-dependent is somewhat acceptable, as it also corroborates previously published data, the statement that BA smooth muscle cell identity depends on the mechanical forces is a speculation and not supported by any evidence. This should be corrected.

10) The penetrance of the phenotype of both klf2a and *notch1b* mutants is very low, so while these factors certainly contribute to OFT valve morphogenesis, the effect is mild. This fact should not be overlooked. What is the effect of the combined loss of klf2a and *notch1b*? n is very low, at least 3 independent experiments should be performed. The font of the graphs is too small and unreadable.

What do the numbers 64% and 56% represent at 48hpf regarding the *notch1b* expression in the manuscript text? From the graph it looks like about 70% of embryos do not show any *notch1b* expression in klf2a mutants at 48hpf. Similar issue is with 72hpf description of the effect. How do the authors define the difference between altered and absent expression? In the examples shown as altered no gene expression can be detected; the exemplary images for all groups should be shown. While loss of klf2a reduces *notch1b* expression it is not completely required.

In the absence of *notch1b*, the expression of klf2 is randomized and not mostly proper as concluded.

The n is too low in the experiments where notch signaling activity is assessed in klf2 mutants. Again at least 3 independent experiments should be performed. Individual data points should be presented. P values for Figures 7C, C' are wrongly written; (10-1 with 1 in superscript). What is the statistical test used?

The images of klf2a reporter expression in *notch1b*^+/-^ and ^-/-^ should be shown.

The previous characterization of the mutants should be properly referenced.

The notch signaling activity seems to be reduced but not gone in the absence of klf2a.

The conclusion that klf2a regulates notch signaling in OFT valve endothelium is an overstatement, and should be corroborated. The data presented are not sufficient to claim this, and are based on assessment of 4 embryos of the reporter that is notoriously variable.

11) How many embryos were used to assess fractional shortening and RFF in the listed mutants? What is the time point for RFS? Individual data points should be plotted. Is the heart rate also unaffected? RFF is a measure for valve function not heart function, this should be corrected or explained better. RFF is assessed at 72 hpf when the valve only starts forming, what is the RFF at 120hpf when the valve leaflets formed?

The assessment of OFT valve morphology is performed in a low number of mutant embryos. Even in controls, 30% of embryos have still thickened valves, and thus the results should not be overstated; in a considerable percentage of the mutants or their combinations, the valves are still formed. The great variability of the phenotypes is especially apparent when the results of A and B are compared.

Therefore it is not evident that Trp and Piezo channels are necessary for the folding of the endothelium; they contribute to some extent to the process.

The data do suggest a redundancy between different channels though. What is the combined effect of trp and piezo channels? Especially the combination of trpp2 and piezo1 would be important to assess.

Where are these channels expressed, in the endothelium only or also in the vascular smooth muscle of the BA?

For the analysis of the Fn1, Elnb, and Yap markers again not sufficient number of embryos is assessed. Fn1 is also expressed in the endothelial cells as the authors show themselves in the Figure 3, and Elnb is a differentiation marker, thus using those two to conclude anything about the identity of the smooth muscle cells of the BA is not appropriate. The concerns here are similar as in the data presented in the Figure 6.

How do these markers look in trpp2 mutants?

The results of the experiments using klf2a reporter in piezo1 and trpp2 mutants are intriguing as they point out to the opposing effects of mechanosensors in regulating klf2a transcriptional activity.

There is not enough evidence that Piezo1 regulates the identity of smooth muscle cells of BA. What is the effect of loss of piezo1 on Yap reporter. Mere presence or absence of Yap in the cells is not sufficient to claim Piezo's role in Yap signaling.

12) The model in Figure 9 is speculative, and is not supported by enough corroborating evidence. The origin of the valve leaflets is tracked only by the optogenetic photoconversion of Kaede to endothelium, the authors do not trace any other cell type within the valve leaflets. While the presence of the vascular smooth muscle cells might affect the OFT valve morphogenesis, the authors do not test any active role of these cells in the valve formation. The role of mechanosensors of Trp an Piezo family in the process of OFT formation is overstated. The evidence points to the modulatory role; they certainly contribute but are not completely necessary for the process of OFT valve formation. There is not enough evidence presented that the mechanosensors play any role in the modulating cell fate or identity of the vascular smooth muscle cells of the BA. The Klf2a regulation of *notch1b* expression and its signaling activity in the valve endothelium is not supported by enough evidence and should be corroborated by functional assays; as presented now it is only a feasible hypothesis.

[Editors' note: further revisions were requested prior to acceptance, as described below.]

Thank you for resubmitting your work entitled "Mechanically activated Piezo channel modulate outflow tract valve development through Yap1 and Klf2-Notch signaling axis" for further consideration at *eLife*. Your revised article has been favorably evaluated by Didier Stainier as the Senior Editor, a Reviewing Editor, and two reviewers.

They praise your efforts and find the revised manuscript substantially improved. The critical issues have been adequately addressed and the reviewers believe the work will receive considerable attention in the field. They however identified a few remaining points, as outlined below:

The introductory text on the architecture of the valves, regarding tricuspid or bicuspid nature, needs to be corrected as detailed in the comments. Further referring to the initial question of robustness and sample size, please add this information to either the main text or figure legends throughout the manuscript.

Reviewer #2:

In the revised version of the manuscript, Duchemin et al. have addressed a majority of the criticisms from all reviewers. In particular, the increase in sample sizes for individual experiments is laudable. The finding that Fn1 and ElnB expression should be coupled with functional analysis should receive attention in the field as well.

Remaining points:

* Sample size: while the authors' issues with increasing the sample numbers due to embryo identification, the involved genotyping, etc. are understandable, stating that other papers did use similar numbers is not really a valid argument. Especially considering *eLife*'s emphasis on transparency and reproducibility, two repetitions of an experiment seem less than ideal. The authors should clearly state the full number of repetitions, etc. throughout the manuscript so the reader can make up her/his own mind about the numbers.

* Please check the issue with bicuspid/tricuspid valve architecture, as also raised by reviewer 3.

* The added schematic in Figure 1A is highly helpful. However, it has a remarkable resemblance (down to individual trabeculae shapes, color scheme, and labeling) to the schematic provided in Felker et al., 2018. This work should be at least referenced in the manuscript.

Reviewer #3:

The authors addressed most of the points raised by all reviewers satisfactorily.

There are few remaining issues that need to be revised.

---

## [Author Response]

Reviewer #1:[…] If I was to point out a weakness of the work, it would be the lack of commenting and possibly analysing the crosstalk between the smooth muscle cells and the endothelium in the process. The final concluding schematic places Piezo1 upstream of different components in both cell compartments, but given that the downstream effectors of Yap1 include modulators of matrix, Bmp and tgfb signalling etc., which will impact on mechanical properties of both endothelium and the smooth muscle cells, and that Notch activity also regulates similar targets, it would seem justified to comment on potential paracrine interactions between the two. I realize however that a full investigation here would go beyond the scope of the current work.

We agree with the reviewer that this point is interesting and deserves further investigation. It is our plan to study this further in the near future, in particular, the interplays between the mechanical properties of both the endothelium and the smooth muscle cells. We agree that this would go beyond the scope of the current work but we now comment on the potential paracrine interactions between the two tissues in the Discussion.

A second aspect that deserves a comment would be how Piezo1 inhibits klf2a whilst Trpp2 is deemed a positive regulator. How is this dichotomy achieved?

This is another interesting observation of our study. While we can just speculate at this point, this can be attributed to the fact that *piezo* and *trpp2* may act in different tissues and/or activate the release of different paracrine factors. To assess if the localization of *piezo1* and *trpp2* mRNA could explain the differential function of these channels, we performed RNAscope assay at 72hpf (Figure 8—figure supplement 3A). We found that *trpp2* is ubiquitously expressed in the embryo, including in the different layers composing the OFT. Similarly, we found that *piezo1* is expressed in both endothelium and smooth muscles, albeit at a lower level than *trpp2*. To confirm these results, we generated a transgenic reporter line with 3kb of the *piezo1* promoter upstream of the start codon (*piezo1:nls-Venus*). We observed the expression of the reporter line mostly in smooth muscles at 72hpf (Figure 8—figure supplement 3B) and cells of the endothelium of the OFT valve. Trpp2 immunohistochemistry showed that Trpp2 is expressed in the endothelium and the smooth muscles confirming that *trpp2* is ubiquitously expressed in the OFT (arrow in Figure 8—figure supplement 3B). We agree that this point deserves more work to be clear. For clarity, we simplified the working model shown in Figure 9 since trpp2 and piezo1 may be active in both cell layers.

Finally, the kdrl:nls-mcherry line appears to label only very few endothelial cells in the posterior OFT and the images here do not really match the schematic. I assume the authors use the fli:lifeact GFP signal to delineate those kdrl negative cells, but this would seem important to comment for technical reasons. On a mechanistic level, it would be important to comment on this cellular heterogeneity in the posterior OFT?

We used the double transgenic line *fli1a:lifect-EGFP; kdrl:nls-mCherry* at 72hpf and 96hpf to show that the *kdrl:nls-mCherry* transgenic line indeed highlights all nuclei of the endothelium, albeit not all nuclei with the same fluorescence intensity. You can see in Author response image 1 the labeling of the OFT endothelium by the line *fli1a:lifect-EGFP; kdrl:nls-mCherry*. Arrowheads in A show cells that express less *kdrl* transgene. To assess if the heterogeneity in expression is also visible at the mRNA level, we performed RNAscope assay at 72hpf using a *kdrl* probe (B). We found that *kdrl* expression might also be heterogenous at the mRNA level. This point deserves more in-depth and proper quantitative analysis to make strong conclusions about the heterogeneity of *kdrl* expression and is beyond the scope of this study. Nevertheless, even though we are not sure if this *kdrl:nlsmCherry* line may display different expression levels because of the transgene insertion, it is possible that *kdrl* expression is heterogeneous in endothelial cells of the OFT.

**Author response image 1. respfig1:** (**A**) Z-section of the kdrl:nls-mCherry and fli1a:lifeact-eGFP in the OFT (**B**) Z-projection of the RNAscope of kdrl in the head and in the OFT (n=6 embryos).

Reviewer #2:[…] At times, the manuscript makes strong conclusions based on small sample sizes (i.e. total embryos analyzed, see below) and the expression of isolated marker genes. Several mutant phenotypes have highly variable penetrance and expressivity (i.e. yap1 mutants, subsection “Klf2, Notch signaling, and Hippo pathways are active in the OFT in different cell layers and are all necessary for proper OFT valve development”), the analysis of which is again hampered by low sample sizes and ambiguous description as to how mutants are identified (assumed Mendelian ratio? Genotyping of individual embryos? etc.). Several statements of necessity and "dramatic" phenotypes need reassessment, as the phenotypes rather indicate the studied genes contribute to robust OFT valve formation rather than being necessary (i.e. notch1b, klf2; see also below). Further, OFT valve formation is a somewhat specialized developmental process, yet the authors seem tempted to draw broad-stroke generalizations about mechanosensing from their findings in the Discussion.* The authors analyze several mutants and state necessity and importance for valve formation. Overall, however, the phenotype analysis is plagued by several issues as presented, and rather suggests the studied factors might be at times dispensable yet contribute to robust valve formation.In all figures with box diagrams, at times vast variability is captured (i.e. Figure 8G'), yet individual data points not shown. Individual data points should be added.

We apologize for this oversight. We have now added the individual data points on the graphs for all figures with data points missing: Figure 7C’, Figure 8F’ and Figure 8G’.

Further, the overall sample size in the majority of the mutant analyses seem low and spotty in the reported details, is shown in percent that are misleading in small sample sizes, and affects the phenotype interpretation. Percentage is used with exceedingly small sample sizes and becomes misleading in significance. Similarly for p-values, that are not sufficient to cover vastly incomplete penetrance of phenotypes. As examples:a) The yap1 mutants have a seemingly increasing penetrance and expressivity of OFT valve phenotypes – yet the phenotype is reported for only 12 embryos (also, are these always the same 12 embryos?), resulting in 58% (6 or 7) of embryos with no valves at 120 hpf; that hardly seems to justify an "important role" but rather a contribution to robust valve formation.

N=12 corresponds to 2 independent experiments. These embryos were analysed and followed over time (i.e. at 72hpf, 96 hpf, 120hpf), so it is n=12 for each time point. These experiments are performed in blind and require that each embryo is genotyped retrospectively, once the imaging has been completed. These embryo numbers are close to what has been published in the recent literature in other zebrafish cardiac studies recently published in *eLife*: See Guerra et al., 2018: Figure 5, n=17, n=12, n=7 or see Nguyen-Chi et al., 2015: Figure 1, n=12, n=3, n=11 or see Semmelhack et al., 2014: Figure 2, n=9).

We changed the text explanation and modified the sentence from “important” to “involved”.

To better quantify the penetrance of the *yap1* mutation, we now show both *yap1^+/+^* and *yap1^+/-^* data in Figure 5B. Indeed, this shows that the *yap1^+/-^*have an intermediate phenotype compared to *yap1^+/+^*and *yap1^-/-^.*

b) Mutant notch1b embryos have a 30% incidence for valve issues, also not warranting the moniker "necessary" but rather contributing. Similarly, the authors state "most" notch1b embryos show proper klf2 expression, yet it's about half at only n=4 (Figure 7).

In order to confirm that *klf2* expression is not affected by *notch1b*, we performed additional experiments using *notch1b^+/+^* and *notch1b^-/-^* embryos in the *klf2a:H2B-GFP* reporter line background. As previously done, we quantified the *klf2a* reporter expression in *notch1b^+/+^* and *notch1b^-/-^*embryos (n=8 and n=6 respectively now) at 72hpf, 96hpf and 120hpf (Figure 7—figure supplement 1). This did not change the final conclusion that we previously stated: *notch1b* mutation does not affect *klf2a* expression.

In addition, we added the confocal images for the *klf2a* reporter and the *kdrl:nls-mCherry* in *notch1b^+/+^* and *notch1b^-/-^* at 72hpf, 96hpf, and 120hpf to display individual data points (Figure 7—figure supplement 1).

c) The authors state that a "large fraction" of piezo1 embryos have less Yap1 at n=7/10, again seeming like an overstatement (can also be called "a bit more than half"?).

We changed a ‘large fraction’ to ‘a bit more than two thirds’ in the text to be more precise.

Such examples go throughout all mutant analysis (also trpp2, etc.) and should be re-phrased and possibly remedied by increasing the sample size to achieve better insights into variability of the phenotypes.

We performed additional immunostaining of Elnb, Fn1, Yap1 in *piezo1^+/+^, piezo1^-/-^, tnnt2a*MO normal beating, and *tnnt2a*MO slow beatingembryos to increase the n and reach 3 independent experiments. These stainings correspond to data on Figure 6D and Figure 8D. Please find in Author response table 1 a recapitulative table showing the numbers for the initial submission and the revised version.

ElnbFn1Yap1Initial submissionRevised versionInitial submissionRevised versionInitial submissionRevised version*piezo1^+/+^*4/411/1411/1216/2010/1010/10*piezo1^-/-^*3/411/159/1220/247/1010/14*tnnt2a*MO normal7/719/224/47/117/711/14*tnnt2a*MO slow5/719/266/614/144/710/16

Author response table 1: table summarizing the additional immunostainings performed for each mutant in comparison to the previous submission.

Also, how do yap1, piezo, trpp2, etc. mutants look like overall? A description of other phenotypes (i.e. cardiac edema, endothelial problems, viability, etc.) would greatly help to gauge the non-autonomous impact of the described defects (or their dependence thereof).

Some overall phenotypes were previously described: *yap1* in Agarwala et al., 2015; *trpp2* in Schottenfeld J et al., 2007 and *trpv4* in Heckel et al. 2014. In addition, we imaged control and mutant embryos (see Author response table 2) and evaluated their overall shape, heart structure, heartbeat and endothelial phenotype (blood flow in the ISV). We also provided pictures of the overall embryo shape and heart structure.

Overall shapeHeart shapeBeatingEndothelial problemsnormalabnormalnormalSitus inversusedemaNormal or abnormalBlood flow in ISV*piezo1^+/+^*10/100/1010/100/100/107/7 normalNormal vasculature See Shmukler et al., 2015 See Shmukler et al., 2016 See Kok et al., 2016*piezo1^-/-^*11/143/14 (tail bent up)13/140/141/1412/12 normal*trpp2+/+, +/-*
18/180/1816/182/180/1818/18 normalNormal vasculature See Goetz et al., 2014 Schottenfeld J et al., 2007*trpp2^-/-^*0/1616/16 (tail bent up)10/166/160/1610/10 normal*trpv4^+/+^*4/62/6 (tail bent up)7/70/70/77/7 normalNormal vasculature (4/4)*trpv4^-/-^*4/84/8 (tail bent up)6/82/80/84/7 normal 3/7 slowNormal vasculature no flow (2/4)*notch1b^+/+^*10/100/109/100/101/1010/10 normalNormal vasculature (21/22)*notch1b^-/-^*10/100/1025/261/260/2610/10 normalNormal vasculature (5/6)*klf2a^+/+^*50/500/5010/100/100/1010/10 normalNormal vasculature (5/5)*klf2a^-/-^*10/100/1010/211/2111/219/10 normal 1/10 slowNormal vasculature (4/5) 1/5 no flow (1/5)*tnnt2a*MO normal17/170/1715/172/170/1710/10 normalNormal vasculature (5/5)*tnnt2a*MO slow7/158/15 (tail bent up)0/150/1515/156/6 slowno flow (5/5)*yap1+/+*
17/170/1717/170/170/1717/17 normalSubtle vascular defects See Nakajima et al., 2017*yap1-/-*
27/281/28 (tail bent up)25/280/283/2826/28 normal, 2/28 no beating

Normal beating: 2-3 Hz

Slow beating: <2 Hz

No beating: 0 Hz

Author response table 2: table summarizing the overall phenotype observed for each mutant.

**Author response image 2. respfig2:** (**A**) Pictures of the different mutants and their controls showing overall shape and heart shape.

* The authors conclude that loss of piezo function impacts smooth muscle "identity" of the BA based on downregulated Fn1 and Elnb staining. While interesting, Fn1 and Elnb are functional/differentiation markers and not necessarily determinants of fate/identity (i.e. Moriyama et al., 2016), yet these observations might rather indicate that flow has a functional contribution to BA maturation. The authors are encouraged to perform a functional BA assay, i.e. active NO metabolism using DAF-2DA (Grimes et al., 2006) to underline the functional impact on BA formation. Further, the claim of Fn1 expression in smooth muscle is not well-funded based on the provided images in Figure 3 and should be shown more clearly.

To clarify the expression domain of Fn1, we used DAF-FMDA assay (similar to the DAF-2DA assay, it reveals active NO metabolism) on wild-type embryos counterstained with Fn1 antibody. These co-stainings revealed the presence of DAF-FMDA activity and Fn1 in the same cells of the OFT at 72hpf, 96hpf and 120hpf, except for the Fn1 staining within the valves. These data are now provided in Figure 3—figure supplement 1. We conclude that Fn1 is localized in smooth muscle cells of the OFT.

For clarity, we also changed the figure panel to provide inverted LUT images of Fn1 and Elnb to make this clearer (Figure 3B, C, D, B’, C’, D’).

To assess the functional impact of mechanical forces on BA formation, we performed the DAFFMDA assay on the *tnnt2a*-morpholino injected embryos and *piezo1* mutants. We also used this assay to measure the BA diameter. In *tnnt2a*-morpholino injected embryos with slow beating (n=6), we observed a smaller BA (diameter=36,1µm ± 4,5, p=10^-3^) and lower fluorescence intensity (intensity=1,4 ± 0,4, p<0,05) than in the *tnnt2a*-morpholino injected embryos displaying normal heartbeat (n=7, BA diameter=47,4µm ± 3,7 and intensity=3,0 ± 1,6). By contrast, the BA size as well as the fluorescence intensity in the *piezo1^-/-^* embryos (n=8, BA diameter=50,5µm ± 4,1, intensity=2,7 ± 1,5) were not significantly different from the *piezo1^+/+^* (n=7, BA diameter=51,3µm ± 3,9, intensity=2,7 ± 1,4) measurements. We conclude that heart activity is required for BA maturation but not *piezo1*, even though *piezo* mutant display abnormal Fn1 and Elnb expression. We added these data to Figure 6E and Figure 8D and Figure 8—figure supplement 2B. As suggested by the reviewer, the results obtained in *piezo* mutants suggest that Fn1 and Elnb may indeed not be functional/differentiation markers and are not necessarily determinants of smooth muscle cell identity. We changed the text of our manuscript accordingly.

Reviewer #3:[…] While the topic of the manuscript is very interesting and timely, the conclusions drawn from the data are not based on enough evidence.Throughout, the number of independent experiments should be clearly stated; the n in most of the experiments is very low. This makes any rigorous assessment futile and greatly undermines the quality of the manuscript.All the quantification/bar graphs should show individual data points wherever possible. The statistical tests should be clearly stated in the figure legends. The manuscript text would benefit from revision; e.g. the last sentence of the Introduction is not clear, or adjectives as reversing flow or tissular remodeling are used and are not correct English.1) In the Introduction: "The multichambered heart contains two different set of valves: arterial valves that are semilunar and mitral (atrioventricular) valves that are tricuspid." Where is the reference to that? Mitral valve is also called the left atrioventricular valve. Tricuspid valve is also called the right atrioventricular valve. Only mitral valve is bicuspid, all other valves are tricuspid in 4 chambered heart. In zebrafish all valves are bicuspid, they are at IFT, AVJ, and OFT. The authors do not mention IFT valve for some reason at all. The valve anatomy in 4 vs 2 chambered heart should be corrected and properly referenced; a simple schematic would be helpful for clarity.

We agree with the reviewer and clarified these issues in the text. In addition, we generated a drawing of the heart and OFT structures in Figure 1.

2) "… the developmental programs driving mitral valve…,less is known about arterial valves" The reference Wu et al. cites more than 20 genes involved in arterial valve development. The rest of the Introduction mentions several signaling pathways involved in aortic valve formation. So meanwhile it is not entirely true that less is known about arterial valves. The Introduction would benefit if it is shortened and refocused on the current knowledge of the effect of mechanical forces on valvulogenesis.

We shortened the Introduction to refocus it on mechanical forces during valvulogenesis.

3) Figure 1A-C: in the text a different transgenic is referred to then shown in the panels. What is the time point in Figure 1A?

We apologize for the misunderstanding and clarified the figure panels. Figure 1A now shows the overall heart structure using the *kdrl:nls-mCherry* (highlights the endocardium) and *myl7:GFP* (highlights the myocardium). Figure 1B is based on the combination of the *fli:lifeacteGFP* and *kdrl:nls-mCherry* lines to highlight the actin and nuclei within the OFT, while Figure 1C uses the combination of the *kdrl:eGFP* and *gata1:dsRed* line for assessing the flow profile. The time point in A is 96hpf and was added on the figure.

4) Figure 2B: in the third panel, there are photoconverted cells anteriorly and posteriorly, where is the anterior cell located at 120 hpf? The photoconverted cells should be highlighted by a star in all panels, not just in one.Consecutive panels for at least 3 embryos should be shown in the supplement.

The cell located anteriorly goes out of frame. This is now indicated in the Legend. Asterisks are included on each panel. We added a Figure 2—figure supplement 1A showing 3 photoconverted embryos for each position (top, middle and bottom).

5) The claim that AVC valve is delaminating should be referenced. Scherz et al., 2008, showed that in zebrafish AVC valve is forming by invagination and not via endocardial cushion formation, similarly to the process of OFT formation that the authors observe here. The conclusion that the OFT valve morphogenesis is unique is thus overstated. It is not clear what the authors mean when they say "unique".

Thanks for pointing that out. Pestel et al., 2016; Grunewald et al., 2018 and Steed et al., 2016 have recently described AVC valve morphogenesis at cellular resolution and showed that the AVC valve forms via endocardial cushion formation. This has been extensively discussed in different research and review articles (Paolini et al., 2018; Donat et al., 2018, Steed, Boselli et al., 2016) and it is now well accepted in the field that zebrafish AVC valves do not form solely via invagination. This is why we consider OFT valve formation to be different from AVC valve formation. This is now clarified in the text. In addition, we have added a Figure 2—figure supplement 1B where we compare the valve formation in the AVC and OFT.

6) The rationale that invagination of the OFT endothelium might be aided by the adjacent tissue should be introduced more clearly. In Figure 1 and 2 only endothelial contribution is observed and shown, yet the authors claim that the cellular contribution of the valve lead them to the hypothesis of adjacent tissue contribution. What is the model?

We agree that this is an exciting aspect of our findings. At this point we can only speculate about the contribution of the smooth muscle cells and endothelial cells in the morphogenetic process. We are currently establishing protocols that allow us to follow OFT valve morphogenesis though time-lapse analysis. We hope that once established, live imaging will provide us with a better understanding of the contribution of these cell layers during OFT valve morphogenesis.

In Figure 3, besides merged images, individual channels for Fibronectin, Elastin b, and endothelial marker should be shown. Counterstaining for Fibronectin and Elastin b should be performed together; from the images presented, not all fibronectin positive cells seem to be Elastin b positive, and the schematic is therefore misleading. If panel A' (Figure 3) is a zoom of panel A, and both are scale bar of 10 microns, why is the scale bar in panel A longer than in panel A'? it does not look like much of a zoom either. This should be revised.

We now show the individual channels. Fibronectin and Elastinb antibodies have been raised in the same organisms so it is not possible to perform co-labeling but we agree that it is possible that not all fibronectin positive cells are elastin b positive. We revised the figures and text accordingly.

We added the single channels for Fn1 and Elnb in Figure 3 as well as revised the scale bars from panels A, A’.

The vascular smooth muscle cells of BA of the OFT are accrued to the ventricle after 48 hpf. This process was recently described e.g. by Felker et al., 2018. Even though the OFT valve formation is occurring concomitantly with the formation of BA, these two events should not be equated, as stated in the conclusions describing Figure 3. If the authors wished to compare OFT valve formation to AVC valve formation, they should provide additional panels of AVC valve for comparison or at least briefly recap the events of AVC valve formation.

We agree with the reviewer that a comparison with AVC valve formation requires clarification and we have now added Figure 2—figure supplement 1B comparing the formation of AVC and OFT valves.

7) What is the stage in Figure 4A and B? The reporters should be visualized with the markers for endothelium and elastin b. The channel for fli:nls-mCherry should be shown especially if the GFP/mCherry ratio is used for the quantification in 4D. How is anterior/posterior boundary defined at different stages for the purpose of the quantification in 4D if the endothelium is invaginating in the direction from posterior to anterior as the authors show in Figure 2? This seems to be arbitrary and misleading due to the dynamic nature of the forming valve. The division medial/lateral or luminal might be more appropriate. What are the individual data points in D? the n should be increased for 120hpf for klf2 reporter; n=2 is not enough! The description of how exactly the quantification was performed should be included, as this read-out is used in the subsequent experiments; is the intensity for both leaflets per embryo averaged? From a single section or z-stack?

We corrected the figure legend to include the embryonic stages and included the *kdrl:nlsmcherry* channel in Figure 4—figure supplement 1. We revised the Materials and methods section to explain better how the boundaries are defined and which statistical tests were performed. We also revised the figure legend to show individual data points.

In addition, we performed an additional experiment to analyze the *klf2a* reporter expression in wild-type at 120hpf to reach n=6 embryos (Figure 4D).

Statistical test should be named in the figure legend.

This has now been done.

8) In Figure 5, the GTIIC:d2GFP Yap reporter should be used together with the endothelial and nuclear marker and elastin b; from the images presented it is not clear how the dashed line was drawn in 5B. Furthermore, Yap reporter should be used together with the nuclear marker, and nuclear/cytoplasmic ratio should be quantified. The antibody staining for Yap appears to be cytoplasmic, which does not confer anything about its signaling activity.The description of 5C should be rewritten; "…a significant fraction of yap1 mutants displayed abnormal valves (17% at 72 hpf…", – 17% show normal valves, the text is misleading. The quantification of the phenotypes should be performed from at least 3 independent experiments, so that appropriate statistical test can be performed.

This has now been done.

Yap1 mutants should be properly referenced.

This has now been done.

If the Yap reporter is active in both endothelial cells and smooth muscle cells of BA, the statement that smooth muscle progenitors are likely to play a role in valve morphogenesis is a speculation and should be corrected. That Yap is essential for OFT valve development is an overstatement when 40% of the mutant embryos still form some sort of a valve structure. Is the 5A' zoom of A? the ROI in A does not correspond to A'?

For the reporter quantification, we are sorry for the misunderstanding: this reporter is cytoplasmic as it reveals the transcriptional activation of tead/yap and it is not a localization reporter. The reporter highlights yap activity and we agree that this is not the case for the antibody.

We provided an Elnb staining on *kdrl:membrane-mCherry x 4xGTTIIC:d2GFP* in Figure 5A. The reporter is expressed in some endothelial cells as well as in the smooth muscles.

We rewrote the text so that we are careful about our conclusions on the role of *yap1* during OFT valve formation.

9) The myl7:GFP reporter line should be used to determine the heart morphology upon different tnnt2 MO doses. From the images presented nothing can be concluded about the MO effect, except perhaps a mild edema in B. The MO doses should be indicated either in the figure directly or figure legend. The BA size should be measured under these MO conditions; it is known that cardiac function may affect the BA size. What is the cut off for the heart rate in the "slow-beating" group? Also the authors should be consistent with the naming, use group 1/2 or beating/slow beating, but not both. The sample number of klf2 and notch reporter has to be increased; the reporters, especially the ones using destabilized GFP are very variable, n of 3 with the usual clutches of at least 100 zebrafish embryos under normal breeding conditions is just not acceptable. The activity of klf2 and notch reporter should be assessed at later time point – 120 hpf when the flow effects are more pronounced, assuming that the data presented are at 72 hpf; this information is missing.

We agree with the reviewer that this part should have been clearer: we selected by eye the embryos showing heart looping, but either with normal heartbeat (called “normal heartbeat”, 2-3Hz) or slow heartbeat (less than 2Hz). We now provide videos of hearts from these two categories for the reader to visualize the different phenotypes. Unfortunately, it is not possible to study 120hpf stages as embryos are not viable at this stage.

As suggested by the reviewer, we used the DAF-FMDA staining for measuring BA width in *tnnt2a*MO injected embryos. We found that it is smaller in the low flow embryos. We added this data to Figure 6E. We conclude from these results that mechanical forces are important for the functional maturation of the OFT.

We performed additional experiments imaging the Notch reporter in klf2a^+/+^ and klf2a^-/-^ to reach n=8 and n=8 respectively, 3 independent experiments. Moreover, we added the individual data point on the graph in Figure 7C’.

We also performed additional imaging of the *klf2a* reporter in *notch1b^+/+^* and *notch1b^-/-^* to reach n=8 and n=6 respectively, 3 independent experiments. We now show in addition the *kdrl:nlsmCherry* channel that was used for the normalization of the *klf2a* reporter intensity. These data are in Figure 7—figure supplement 1.

In the experiments assessing the Fb1, Eln B and Yap abundance, both the smooth muscle cells of BA and endothelial cells have to counterstained, e.g. NO fluorescent indicator DAR4-AM is a reliable marker of the BA smooth muscle cells. It is not clear if the smooth muscles cells are present at all with the high dose of tnnt2 MO, and as such the "missing" staining can be simply a result of missing cells. In Figure 6D, at what time point the valve presence was quantified? n=9 for tnnt2 MO is very low; are the authors really injecting just 9 embryos?The controls are completely missing in the experiments presented in Figure 6, with the exception of Figure 6D.

We consider the normal beating heart category as our internal controls for *tnnt2a*MO-injected embryos with slow beating heart from the same clutch. Indeed, the injection of the diluted morpholino leads to various heartbeat phenotypes and we selected embryos with a heartrate of about 2-3 Hz for controls, and the ones with slower heartrate as our experimental condition.

While the conclusion that klf2 and notch activity are flow-dependent is somewhat acceptable, as it also corroborates previously published data, the statement that BA smooth muscle cell identity depends on the mechanical forces is a speculation and not supported by any evidence. This should be corrected.

To address this point, we performed new experiments with DAF-FMDA in *tnnt2a*MO-injected embryos with normal beating heart and slow beating heart to confirm the presence of the smooth muscles. However, they are not as functional as in the controls since DAF-FMDA expression is lower than in “normal beating heart” embryos. We thus conclude that BA smooth muscle cells do not mature properly when mechanical forces are abnormal. We added these data to Figure 6E.

10) The penetrance of the phenotype of both klf2a and notch1b mutants is very low, so while these factors certainly contribute to OFT valve morphogenesis, the effect is mild. This fact should not be overlooked. What is the effect of the combined loss of klf2a and notch1b? n is very low, at least 3 independent experiments should be performed. The font of the graphs is too small and unreadable.

We performed additional experiments imaging the Notch reporter in *klf2a^+/+^* and *klf2a^-/-^* to reach n=8 and n=8 respectively, 3 independent experiments. Moreover, we added the individual data point on the graph in Figure 7C’. We also performed additional imaging of the *klf2a* reporter in *notch1b^+/+^*and *notch1b^-/-^*to reach n=8 and n=6 respectively, 3 independent experiments. We additionally show the *kdrl:nls-mCherry* channel that was used for the normalization of the *klf2a* reporter intensity. These data are in Figure 7—figure supplement 1.

We are also interested to know the effect of the combined *klf2a; notch1b* mutants but unfortunately, we do not have these combined mutants growing in our facility. Generating the compound mutants would require a much longer period of time than that given to us to perform the revisions so we did not perform this experiment. However, we plan to study these mutants in the near future.

What do the numbers 64% and 56% represent at 48hpf regarding the notch1b expression in the manuscript text?

64% corresponds to the% of embryos with altered expression of *notch1b* in the *klf2a^-/-^.* Thanks to the reviewer, we realized that the 56% and the “50%” are not referring to data shown in this figure. We revised the text accordingly.

From the graph it looks like about 70% of embryos do not show any notch1b expression in klf2a mutants at 48hpf. Similar issue is with 72hpf description of the effect. How do the authors define the difference between altered and absent expression? In the examples shown as altered no gene expression can be detected; the exemplary images for all groups should be shown. While loss of klf2a reduces notch1b expression it is not completely required.

‘Altered’ expression means that there is still expression in the heart but the expression pattern is abnormal, mislocalized. ‘Absent’ means that no ISH staining was observed in these embryos. We agree that *klf2a*^-/-^ display a reduction in *notch1b* expression and it is not completely required. We provide representative images in Figure 7—figure supplement 1.

In the absence of notch1b, the expression of klf2 is randomized and not mostly proper as concluded.The n is too low in the experiments where notch signaling activity is assessed in klf2 mutants. Again at least 3 independent experiments should be performed. Individual data points should be presented. P values for Figures 7C, C' are wrongly written; (10-1 with 1 in superscript). What is the statistical test used?

We performed additional experiments imaging the Notch reporter in *klf2a^+/+^* and *klf2a^-/-^* to reach n=8 and n=8 respectively, 3 independent experiments. Moreover, we added the individual data point on the graph in Figure 7C’. We also performed additional imaging of the *klf2a* reporter in *notch1b^+/+^* and *notch1b^-/-^* to reach n=8 and n=6 respectively, 3 independent experiments. We now show in addition the *kdrl:nls-mCherry* channel that was used for the normalization of the *klf2a* reporter intensity. These data are in Figure 7—figure supplement 1. The statistical test used is the Student t-test. This is also included in the Materials and methods section.

The images of klf2a reporter expression in notch1b +/- and -/- should be shown.

We have not considered the *notch1b^+/-^*for that experiment.

The previous characterization of the mutants should be properly referenced.

All the published mutants are now referenced in our revised version.

The notch signaling activity seems to be reduced but not gone in the absence of klf2a.The conclusion that klf2a regulates notch signaling in OFT valve endothelium is an overstatement, and should be corroborated. The data presented are not sufficient to claim this, and are based on assessment of 4 embryos of the reporter that is notoriously variable.

We performed additional experiments imaging the Notch reporter in *klf2a^+/+^* and *klf2a^-/-^*to reach n=8 and n=8 respectively, 3 independent experiments. Moreover, we added the individual data point on the graph in Figure 7C’.

11) How many embryos were used to assess fractional shortening and RFF in the listed mutants? What is the time point for RFS? Individual data points should be plotted. Is the heart rate also unaffected? RFF is a measure for valve function not heart function, this should be corrected or explained better. RFF is assessed at 72 hpf when the valve only starts forming, what is the RFF at 120hpf when the valve leaflets formed?

At least 3 embryos were used for each category for RFF and at least 3 other embryos for the fractional shortening. The embryos were analysed at 72 hpf.

To answer this point, we assessed the RFF at 120hpf. A table for the RFF at 120hpf is shown in Figure 8—figure supplement 1B. We also looked at the heartbeat in all mutants (see Author response table 3). RFFs were not dramatically altered in these mutants suggesting that the valves are still operating in the mutants even though the morphogenetic program is affected.

Overall shapeHeart shapeBeatingEndothelial problemsnormalabnormalnormalSitus inversusedemaNormal or abnormalBlood flow in ISV*piezo1^+/+^*10/100/1010/100/100/107/7 normalNormal vasculature See Shmukler et al., 2015 See Shmukler et al., 2016 See Kok et al., 2016*piezo1^-/-^*11/143/14 (tail bent up)13/140/141/1412/12 normal*trpp2+/+, +/-*
18/180/1816/182/180/1818/18 normalNormal vasculature See Goetz et al., 2014 Schottenfeld J et al., 2007*trpp2^-/-^*0/1616/16 (tail bent up)10/166/160/1610/10 normal*trpv4^+/+^*4/62/6 (tail bent up)7/70/70/77/7 normalNormal vasculature (4/4)*trpv4^-/-^*4/84/8 (tail bent up)6/82/80/84/7 normal 3/7 slowNormal vasculature no flow (2/4)*notch1b^+/+^*10/100/109/100/101/1010/10 normalNormal vasculature (21/22)*notch1b^-/-^*10/100/1025/261/260/2610/10 normalNormal vasculature (5/6)*klf2a^+/+^*50/500/5010/100/100/1010/10 normalNormal vasculature (5/5)*klf2a^-/-^*10/100/1010/211/2111/219/10 normal 1/10 slowNormal vasculature (4/5) 1/5 no flow (1/5)*tnnt2a*MO normal17/170/1715/172/170/1710/10 normalNormal vasculature (5/5)*tnnt2a*MO slow7/158/15 (tail bent up)0/150/1515/156/6 slowno flow (5/5)*yap1+/+*
17/170/1717/170/170/1717/17 normalSubtle vascular defects See Nakajima et al., 2017*yap1-/-*
27/281/28 (tail bent up)25/280/283/2826/28 normal, 2/28 no beating

Normal beating: 2-3 Hz

Slow beating: <2 Hz

No beating: 0 Hz

Author response table 3: table summarizing the overall phenotype observed for each mutant.

The assessment of OFT valve morphology is performed in a low number of mutant embryos. Even in controls, 30% of embryos have still thickened valves, and thus the results should not be overstated; in a considerable percentage of the mutants or their combinations, the valves are still formed. The great variability of the phenotypes is especially apparent when the results of A and B are compared.

Numbers for OFT valve morphology in *trpp2* and *piezo1* mutants are: in *trpp2^+/+^* (n=11, n=10, n=10), *trpp2^-/-^* (n=8, n=13, n=13), piezo1^+/+^ (n=10, n=10, n=9) and *piezo1^-/-^* (n=10, n=10, n=9) at 72hpf, 96hpf and 120 hpf respectively, which are standards for live-imaging. Moreover, these embryos are followed over time.

Therefore it is not evident that Trp and Piezo channels are necessary for the folding of the endothelium; they contribute to some extent to the process.The data do suggest a redundancy between different channels though. What is the combined effect of trp and piezo channels? Especially the combination of trpp2 and piezo1 would be important to assess.

In addition, we analyzed the valve phenotype of *piezo1^-/-^; trpp2* MO to address the redundancy between *trpp2* and *piezo1* channels (Figure 8A, B, C). The *trpp2* MO recapitulates well the *trpp2* mutant phenotype and is well accepted by the community.

Where are these channels expressed, in the endothelium only or also in the vascular smooth muscle of the BA?

*Trpp2* is believed to be ubiquitously expressed. For *piezo1*, we generated a transgenic line expressing venus FP under the 3kb *piezo* promoter. The results suggest that the expression of the *piezo1* reporter is mainly in the smooth muscles.

In addition, we performed Trpp2 antibody staining on our *piezo1* reporter line. Trpp2 seems to be expressed in smooth muscles as well as endothelium whereas *piezo1* seems to be expressed mainly in smooth muscles (Figure 8—figure supplement 3B).

Finally, we performed RNAscope using the *trpp2* and *piezo1* fluorescent probes to better assess the localization of these channels. The results suggest that both channels are expressed in the endothelium and smooth muscles. However, *trpp2* is expressed in many cells of the OFT while *piezo1* is expressed in only a subset of cells in the OFT.

For the analysis of the Fn1, Elnb, and Yap markers again not sufficient number of embryos is assessed. Fn1 is also expressed in the endothelial cells as the authors show themselves in the Figure 3, and Elnb is a differentiation marker, thus using those two to conclude anything about the identity of the smooth muscle cells of the BA is not appropriate. The concerns here are similar as in the data presented in the Figure 6.

We performed additional immunostaining of Elnb, Fn1, Yap1 in *piezo1^+/+^, piezo1^-/-^, tnnt2a*MO normal beating, and *tnnt2a*MO slow beatingembryos to increase the n and reach 3 independent experiments. These stainings correspond to data on Figure 6D and Figure 8D. Author response table 4 is a recapitulative table with the numbers for the initial submission and the revised version.

ElnbFn1Yap1Initial submissionRevised versionInitial submissionRevised versionInitial submissionRevised version*piezo1^+/+^*4/411/1411/1216/2010/1010/10*piezo1^-/-^*3/411/159/1220/247/1010/14*tnnt2a*MO normal7/719/224/47/117/711/14*tnnt2a*MO slow5/719/266/614/144/710/16

Author response table 4: table summarizing the additional immunostainings performed for each mutant by comparison to previous submission.

How do these markers look in trpp2 mutants?

These markers are not affected in the *trpp2* mutants.

The results of the experiments using klf2a reporter in piezo1 and trpp2 mutants are intriguing as they point out to the opposing effects of mechanosensors in regulating klf2a transcriptional activity.There is not enough evidence that Piezo1 regulates the identity of smooth muscle cells of BA. What is the effect of loss of piezo1 on Yap reporter. Mere presence or absence of Yap in the cells is not sufficient to claim Piezo's role in Yap signaling.

We do not know the effect of piezo in YAP reporter. Here we present Yap1 expression analysis using the antibody. However, we did not intend to claim that piezo modulates yap signaling. To clarify the BA phenotype, we provide analysis of the DAF-2DA smooth muscle marker in the *piezo1^+/+^* and *piezo1^-/-^*mutants. The BA size as well as the fluorescence intensity in the *piezo1^-/-^* embryos (n=8, BA diameter=50,5µm ± 4,1, intensity=2,7 ± 1,5) were not significantly different from the *piezo1^+/+^* (n=7, BA diameter=51,3µm ± 3,9, intensity=2,7 ± 1,4) measurements. We added these data to Figure 8.

12) The model in Figure 9 is speculative, and is not supported by enough corroborating evidence. The origin of the valve leaflets is tracked only by the optogenetic photoconversion of Kaede to endothelium, the authors do not trace any other cell type within the valve leaflets. While the presence of the vascular smooth muscle cells might affect the OFT valve morphogenesis, the authors do not test any active role of these cells in the valve formation. The role of mechanosensors of Trp an Piezo family in the process of OFT formation is overstated. The evidence points to the modulatory role; they certainly contribute but are not completely necessary for the process of OFT valve formation. There is not enough evidence presented that the mechanosensors play any role in the modulating cell fate or identity of the vascular smooth muscle cells of the BA. The Klf2a regulation of notch1b expression and its signaling activity in the valve endothelium is not supported by enough evidence and should be corroborated by functional assays; as presented now it is only a feasible hypothesis.

We agree with the reviewer and apologize for the misunderstanding. The model is indeed speculative and is an attempt to provide discussion material. We carefully reassessed the text and discussion to make sure that the reader understands that this is a working model. We agree that mechanosensors have a modulatory role in the process of OFT morphogenesis and changed the title of our manuscript accordingly.

The regulation of *notch1b* by *klf2a* will be reinforced by the experiments performed. We have now reached n=8 and n=8, respectively, and 3 independent experiments.

We also performed additional imaging of the *klf2a* reporter in *notch1b^+/+^* and *notch1b^-/-^* to reach n=8 and n=6 respectively, 3 independent experiments. We now show the *kdrl:nls-mCherry* channel that was used for the normalization of the *klf2a* reporter intensity. These data are in Figure 7—figure supplement 1.

[Editors' note: further revisions were requested prior to acceptance, as described below.]

Reviewer #2:[…] Remaining points:* Sample size: while the authors' issues with increasing the sample numbers due to embryo identification, the involved genotyping, etc. are understandable, stating that other papers did use similar numbers is not really a valid argument. Especially considering eLife's emphasis on transparency and reproducibility, two repetitions of an experiment seem less than ideal. The authors should clearly state the full number of repetitions, etc. throughout the manuscript so the reader can make up her/his own mind about the numbers.

This is now corrected.

* Please check the issue with bicuspid/tricuspid valve architecture, as also raised by reviewer 3.

Sincere apologies for this oversight, this is now corrected.

* The added schematic in Figure 1A is highly helpful. However, it has a remarkable resemblance (down to individual trabeculae shapes, color scheme, and labeling) to the schematic provided in Felker et al., 2018. This work should be at least referenced in the manuscript.

The reviewer is right thanks for pointing this out. This is now corrected.